# Humans in Kitchens: A Dataset for Multi-Person Human Motion Forecasting with Scene Context

**Julian Tanke**[*]
tanke@iai.uni-bonn.de

**Oh-Hun Kwon**[*]
ohkwon@uni-bonn.de

**Felix B Mueller**[*,†]
felix.benjamin.mueller@uni-bonn.de

**Andreas Doering**[*,‡]
doering@iai.uni-bonn.de

**Juergen Gall**[*,‡]
gall@iai.uni-bonn.de

## Abstract

Forecasting human motion of multiple persons is very challenging. It requires to model the interactions between humans and the interactions with objects and the environment. For example, a person might want to make a coffee, but if the coffee machine is already occupied the person will have to wait. These complex relations between scene geometry and persons arise constantly in our daily lives, and models that wish to accurately forecast human behavior will have to take them into consideration. To facilitate research in this direction, we propose Humans in Kitchens, a large-scale multi-person human motion dataset with annotated 3D human poses, scene geometry and activities per person and frame. Our dataset consists of over 7.3h recorded data of up to 16 persons at the same time in four kitchen scenes, with more than 4M annotated human poses, represented by a parametric 3D body model. In addition, dynamic scene geometry and objects like chair or cupboard are annotated per frame. As first benchmarks, we propose two protocols for short-term and long-term human motion forecasting.

## 1  Introduction

Understanding and anticipating human motion within groups is very challenging and essential in the context of socially-compliant autonomous robots [1, 2, 3, 4, 5, 6, 7, 8], as they must possess the ability to understand and respond appropriately to human behavior. Moreover, this topic has relevance in the fields of neuroscience and social sciences [9, 10, 11], as it enables the development of computational models that explore the perception of others' behavior and its influence on one's own behavior. For example, imagine a group of persons sitting on a sofa and another person walking towards it, one would expect the person to sit down on an unoccupied seat on the sofa. Similarly, two persons in front of a whiteboard are expected to discuss their ideas. However, it is more plausible that only one of them writes onto the whiteboard while the other observes.

Capturing social interactions and interactions with the environment necessitates a large dataset for effective training and evaluation. Such dataset must possess three essential characteristics: (a) it should encompass natural interactions between multiple individuals recorded in a real environment,

---

[*]University of Bonn
[†]Fraunhofer Institute for Intelligent Analysis and Information Systems IAIS
[‡]Lamarr Institute for Machine Learning and Artificial Intelligence

37th Conference on Neural Information Processing Systems (NeurIPS 2023) Track on Datasets and Benchmarks.

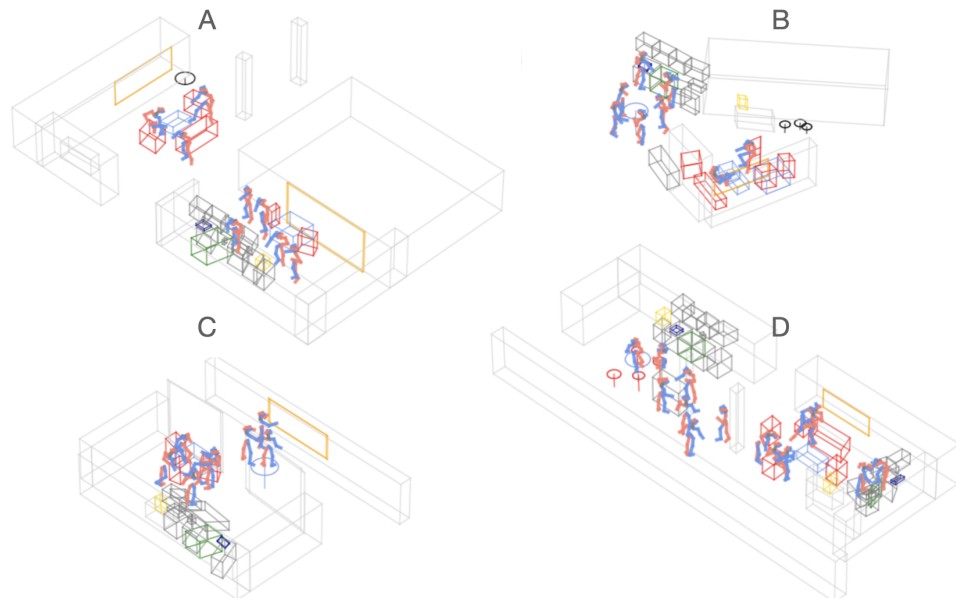

Figure 1: Humans in Kitchens consists of 7.3h captured human poses of multiple persons in four different kitchen environments `A`, `B`, `C` and `D`.

(b) it should have annotated scene geometry to account for interactions with the scene, and (c) it should include annotated per-person action labels to balance the evaluation and to avoid a strong bias towards simple activities like standing, walking, and sitting. Currently, such dataset with 3D human poses does not exist as shown in Tab. 1. The largest multi-person human motion dataset is Panoptic Studio [12]. The dataset, however, has been recorded in a studio. Although the persons interact, they mainly stand due to the small recording area. It also does not include a real environment and the subjects need to act in an unfamiliar environment with many cameras and light sources, which can induce a behaviour that differs from real-life behaviour.

In order to address these issues, we propose *Humans in Kitchens*, a large-scale multi-person 3D human motion dataset with annotated scene geometry and per-person activities. Our dataset consists of more than 4M unique poses of 90 individuals in total. We recorded persons in four real kitchen for over 7.3h. Each of the four kitchen sequences was continuously recorded for 1.5h to 1.9h. Persons could freely enter or leave the scene and received minimal instructions, resulting in a very natural behavior and interactions. For each scene, we annotated objects that people may interact with, such as sinks, dishwashers, chairs, or whiteboards, and objects that determine the geometry of the scene, such as walls. Some objects, such as chairs and kettles, are annotated per frame as they may be moved around the scene. The objects, however, are only coarsely annotated by 3D boxes and cylinders. While such an annotation allows to learn based on scene context where and how activities are performed, the dataset cannot be used to model fine-grained human-object interactions like rotating the knob of the microwave or grasping a knife. The scenes span between $38\mathrm{m}^2$ to $80\mathrm{m}^2$, which is much larger than the $19\mathrm{m}^2$ of Panoptic Studio [12]. The maximum number of individuals in the scene at the same time is 16, twice the maximum number of persons in Panoptic Studio. For each person, we annotate their frame-wise activity, such as *walking*, *sitting*, *writing on whiteboard* or *making coffee*. We represent humans by the SMPL [13] body model, which includes the 3D skeleton pose. We belief that Humans in Kitchens will contribute to advance multi-person human motion forecasting as well as modeling scene context for social behaviour understanding and anticipation.

We provide details on the acquisition and annotation of the dataset, dataset statistics, and evaluate state-of-the-art methods for multi-person human motion forecasting. We further discuss limitations and potential risks of the dataset.

## 2 Related Datasets

We present the most related datasets with one or multiple 3D human poses in Tab. 1 and briefly discuss them.

Table 1: Comparison of various datasets with 3D human poses. The *real data* column specifies if a dataset contains real, synthetic, or partially synthetic human motion. The *real setting* column specifies if the recording was done in a controlled studio environment or in a real-world scene. The *SMPL* column determines whether a dataset provides SMPL [13] poses. The column $\max(\#P)$ specifies the maximum number of persons at the same time in a scene while columns *activities* and *scene* determine if the dataset contains per-frame annotations of activities or scene geometry. Numbers in *activities* indicate the number of different per-frame activities.

| Dataset | real data | real setting | SMPL | $\max(\#P)$ | total time | activities | scene | framerate |
|---|---|---|---|---|---|---|---|---|
| AIST++ [14] | yes | no | yes | 1 | 5.2h | no | no | 60Hz |
| AMASS [15] | yes | no | yes | 1 | 40h | no | no | 60Hz |
| BEHAVE [16] | yes | no | yes | 1 | 8.5min | no | yes | 10Hz |
| CHAIR [17] | yes | no | yes | 1 | 17.3h | no | yes | 30Hz |
| CHICO [18] | yes | yes | no | 1 | 3.77h | no | yes | 25Hz |
| GIMO [19] | yes | yes | yes | 1 | 1.2h | no | yes | 30Hz |
| GTA-IM [20] | no | no | no | 1 | 9.2h | no | yes | 30Hz |
| Human3.6M [21] | yes | no | no | 1 | 2.93h | no | no | 50Hz |
| Humanise [22] | no | no | yes | 1 | 5.55h | yes (language) | yes | 60Hz |
| MoGaze [23] | yes | no | no | 1 | 3h | no | yes | 120Hz |
| PROX [24] | yes | yes | yes | 1 | 55min | no | yes | 30Hz |
| SAMP [25] | partial | no | yes | 1 | 100min | no | yes | 30Hz |
| 3DPW [26] | yes | yes | yes | 2 | 14min | no | no | 60Hz |
| CHI3D [27] | yes | yes | no | 2 | 40min | no | no | 200Hz |
| CMU Mocap [28] | yes | no | no | 2 | 9.75h | no | no | 60Hz / 120Hz |
| CMU Panoptic [29] | yes | no | no | 8 | 5.5h | no | no | 29.97Hz |
| EgoBody [30] | yes | yes | yes | 2 | 2h | no | yes | 30Hz |
| ExPI [31] | yes | no | no | 2 | 20min | no | no | 25Hz |
| Haggling dataset [32] | yes | no | no | 3 | 3h | yes (1) | no | 29.97Hz |
| MuPoTS-3D [33] | yes | yes | no | 3 | ≤4.4min | no | no | 30Hz / 60Hz |
| NTU-RGB+D 120 [34] | yes | no | no | 2 | 63min | yes (120) | no | 30Hz |
| RICH [35] | yes | yes | yes | 2 | 2.7h | no | yes | 60Hz |
| Humans in Kitchens (ours) | yes | yes | yes | 16 | 7.33h | yes (82) | yes | 25Hz |

**Single-Person Datasets**: AMASS [15] unifies several 3D human motion datasets using SMPL [13]. In total, AMASS contains over $40$ hours of motion capture recordings. While the underlying articulated model ensures high quality motion, AMASS only contains recordings of a single person and thus no human interactions. It also does not contain scene context information or per-frame action annotations. BEHAVE [16] contains fine-grained human-object interactions but it is recorded at only 10Hz and very small (8.5 minutes). Another human-object interaction dataset is CHICO [18], which contains 3.77 hours of recording. In contrast to BEHAVE, the human poses are represented by 3D keypoints instead of SMPL [13] body poses. Human3.6M [21] contains around 900k high-quality human 3D poses. The motion sequences, however, are unrealistic since the actors pantomime activities without objects except of sitting on a chair. Other objects or 3D geometry are missing. MoGaze [23] contains 3 hours of human-object interactions with annotated 3D geometry. PROX [24] provides very high-quality human-object interactions with static objects such as chairs, beds and sofas and utilizes SMPL as body model. POSA [36] extends this with a generative model. The GTA Indoor Motion dataset (GTA-IM) [20] utilizes the GTA engine to produce human-scene interactions in indoor environments for human motion forecasting. The motion in this dataset, however, is synthesized and looks unrealistic and clumsy. Similarly, SynBody [37] utilizes pre-rendered scenes and synthetic humans to produce diverse sequences. SAMP [25] is a human-scene interaction dataset containing 7 real objects, such as sofas and armchairs, and various human interactions with those. An extensive augmentation pipeline is used to extend the dataset with a greater variety of human-object interactions. AIST++ [14] contains 30 subjects dancing to music sequences. GIMO [19] contains a single person interacting with a static, high-quality mesh, where data is provided in the form of an egocentric viewpoint. The dataset was explicitly designed for motion forecasting, where a person walks into a scene with the intent to interact with an object. Our dataset contains multiple persons, 5 times more frames, and moving objects. Humanise [22] is a large-scale synthetic human-object interaction dataset that leverages existing datasets in human motion (AMASS) and 3D indoor scenes [38]. Similar to GIMO, the scenes are static. Furthermore, the person-object interactions are synthetic and do not include real person-object interactions. CHAIR [17] is a large-scale human-chair interaction dataset that contains a large variation of chairs. In contrast to these datasets, Humans in Kitchens contains multiple persons.

**Multi-Person Datasets**: 3DPW [26] contains 2 persons per scene, but the 3D joint locations were obtained from moving cameras, resulting in unrealistic sliding. In total, this dataset consists of 14 minutes of recording. CMU-Mocap [28] is a high-quality motion capture dataset of 1 to 2 persons. While some of the single-person sequences contain scene interaction, the scenes are not annotated. The dataset does not contain accurate action labels, but the sequences can be searched based on

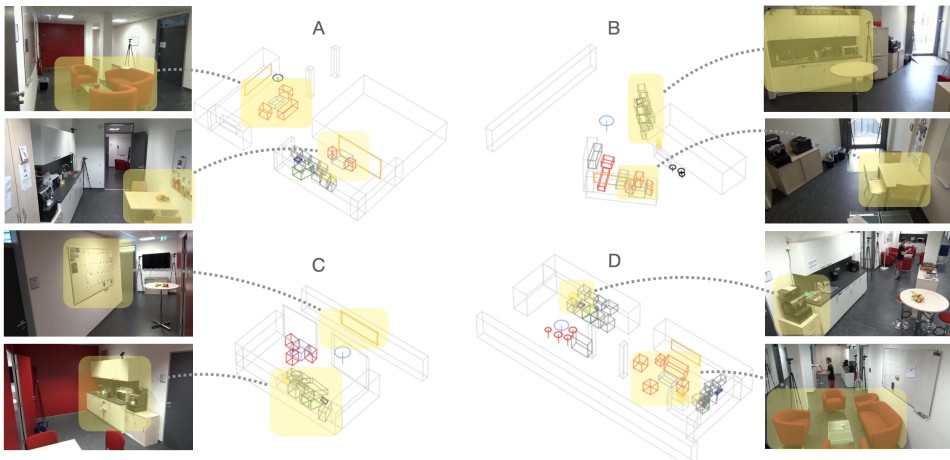

Figure 2: Overview of each of the four kitchens. Each kitchen contains sofas and chairs (red ■), tables (blue ■) and at least one whiteboard (orange ■), fridge (green ■), coffee machine (yellow ■) and sink (dark blue ■). Best viewed using zoom in PDF viewer.

the video descriptions. CHI3D [27] consists of 40 minutes of two-person interactions, including close interactions such as touching. Human poses are represented as 3D skeletons and as SMPL-X models [39]. CMU Panoptic Studio [12, 29] is a large-scale 3D human motion dataset featuring 1 to 8 persons in various scenes. The human poses are represented using 3D COCO keypoints. The dataset size is 5.5h, but lacks labeled scene geometry and per person action annotations. Due to the studio recording, the range of motion is limited. MuPoTS-3D [33] is a test set for 3D human pose estimation. It contains up to 3 persons in a scene but it contains only 4.4 minutes of recording. NTU-RGB+D [40, 34] is an action recognition dataset containing more than 1 hour of recording of one or two persons. Human bodies are represented as 3D skeletons and per-frame activities are annotated. In contrast to these datasets, Humans in Kitchens has been captured in a real environment, includes annotated scene context and frame-wise activities per person. EgoBody [30] contains sequences of pairwise social interactions with real scene geometry. In contrast to our work, the scene geometry is only statically annotated and social interactions are less natural, as one actor wears a virtual headset. RICH [35] captures human-object interactions by defining pseudo-contact labels on the body mesh. Unlike PROX [24], RICH contains mostly outdoor scenes of around $60m^2$ and provides more accurate SMPL-X estimates. Some of the scenes contain two humans who interact with each other from a distance like throwing a ball.

## 3 Humans in Kitchens

In order to obtain a dataset that on one hand contains realistic behaviour of multiple interacting persons and on the other hand is GDPR conform, which includes the informed consent of each subject in the dataset, we followed a different approach than previous datasets with 3D human poses. Instead of asking subjects to perform certain motions or games in a recording studio, we collected data in four real office kitchens for a duration between 1.5h and 1.9h. Persons were are allowed to enter and leave the scene and persons that did not want to participate were asked to use another kitchen during the recording session. Despite of the informed consent sheet, only a subset of the participants received minimal instructions as we will discuss in Sec. 3.1.1. The dataset acquisition and annotation process will be described in Sec. 3.1 and Sec. 3.2.

### 3.1 Dataset Acquisition

For the data acquisition, 11-12 calibrated cameras that were synchronized via the audio signal have been used. The recording has been performed in four different office kitchens on different days. The layout of the kitchens is shown in Fig. 2. While the hardware setup is described in [41], the corresponding repository [41] contains only the raw data where the estimated 3D human poses are very noisy due to severe occlusions and limited view of each camera. The raw data itself can thus not be used for training or evaluation. In Sec. 3.1.2, we describe how the raw data has been manually

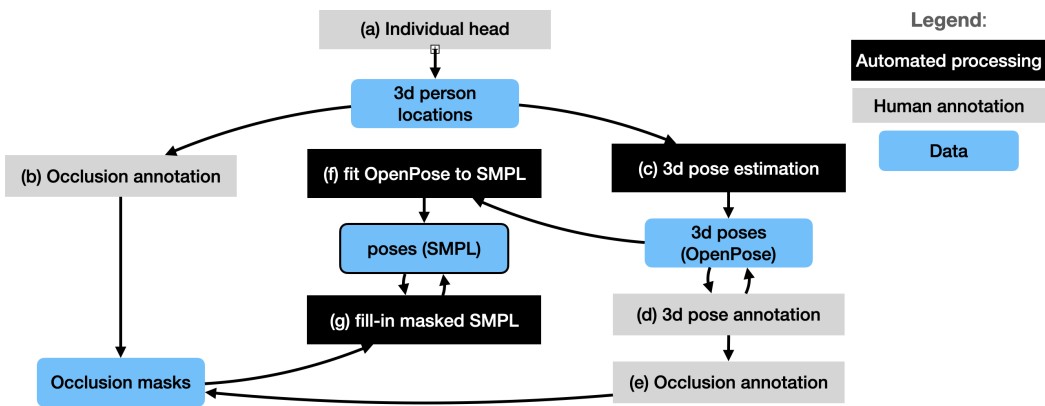

Figure 3: Overview of the 3D human pose annotation process. First, (a) the heads of individual persons are annotated in each frame, represented as a 3D point in global coordinates and unique identity. Given the person annotations, we annotate occlusions for each person and frame (b) and automatically extract 3D human poses (c). We manually correct 3D human poses (d) or mask them (e) if even the annotators cannot correct them. We estimate SMPL parameters (f) and inpaint masked regions (g) to obtain for each pose a full SMPL pose representation. Black boxes represent automated processes without human intervention, gray boxes represent human annotation processes and blue boxes represent data.

annotated to obtain the Humans in Kitchens dataset for multi-person human motion forecasting with scene context.

### 3.1.1 Behavior Protocol

For each of the four recordings, a cake has been provided to attract participants to the kitchen. To facilitate behavior as natural as possible, we provided only minimal instructions to 10 persons in each recording, where they were asked to randomly perform 3 of the following activities at any time and in any order:

**Make coffee**: Prepare a coffee and drink it; **Make tea**: Use the kettle to prepare a tea and drink it; **Eat cake**: Take a slice of the cake and eat it; **Eat fruit**: Eat some of the provided fruits; **Drink water**: Drink water from the tap; **Explain on Whiteboard**: Explain a topic of your choice on the whiteboard; **Use Laptop**: Work on a laptop; **Use Microwave**: Use the microwave to heat milk for coffee; **Read paper**: Read a paper; **Make a phone call**: Make a phone call; **Clean dish**: Clean your dish(es) in the sink; **Place in dishwasher**: Put used dishes in the dishwasher.

While 10 persons were instructed, the other persons present in the scene were not instructed to perform any of the above mentioned activities. However, each person was allowed to perform any activity, e.g., anyone could make a coffee, eat a cake or clean dishes.

### 3.1.2 Pose and Activity Annotation

We annotated the 3D human poses of each person in each frame where each person has a unique identity through the sequence. To extract 3D human poses in our very challenging environment, we annotated the 3D poses in five phases:

**Manual nose annotations**: We first manually annotated each individual in each frame at their nose in the 3D scene, using a custom annotation tool. This also allowed us to re-identify persons who left the scene but later returned to the recording. We verified the correct annotations in a second pass where additionally the 82 activities where annotated per frame per person. In total, annotating this phase took around 2,000 person hours.

**Automated human pose estimation**: We ran an off-the-shelf 3D human pose estimation method [42] to extract the 3D human poses from the multiple cameras. We match the estimated 3D poses to the closest manually annotated nose and drop all leftover poses. If a 3D nose annotation is not matched to an estimated 3D pose, we linearly interpolate between the previous and the next frame. The extracted 3D human poses are represented as 3D skeletons using the OpenPose keypoints [43].

Table 2: Dataset statistics for the four kitchen environments A, B, C and D.

| | A | B | C | D | total |
|---|---|---|---|---|---|
| # frames | 128,959 (1.43h) | 179,097 (1.99h) | 175,392 (1.95h) | 176,264 (1.96h) | 659,712 (7.33h) |
| # annotated poses | 573,253 | 1,132,422 | 908,380 | 1,415,189 | 4,029,244 (44.76h) |
| mean persons / frame | 4.42 | 6.32 | 5.17 | 7.97 | - |
| median persons / frame | 4 | 6 | 5 | 7 | - |
| max. persons / frame | 9 | 14 | 9 | 16 | - |
| # individuals | 18 | 32 | 16 | 24 | 90 |
| surface area | $76.35m^2$ | $57.22m^2$ | $38.28m^2$ | $80.40m^2$ | - |
| # camera views | 11 | 11 | 12 | 12 | - |
| # scene objects | 37 | 40 | 29 | 50 | - |

**Manual occlusion masking and human pose correction**: The automated pose estimation method fails in heavily occluded scenes, requiring us to manually correct the 3D skeletons in those frames. We manually annotated and corrected 20,000 3D poses, around 0.5% of the dataset. Additionally, we manually annotated occlusion masks for head, upper body and lower body where even human annotators were not able to determine the correct 3D joint positions. Differentiating between head, upper and lower body is a compromise between accuracy and annotation speed, as often only certain parts of a person, e.g., the legs, were occluded. In total, 6% of the poses have at least one body part masked. This annotation phase required 600 person hours.

**Fitting SMPL**: We adapt an off-the-shelf optimization framework[4] to extract SMPL parameters from the 3D OpenPose keypoints that have been annotated in the previous steps. We extract the SMPL parameters for all poses, even the masked once. We can do this as the masked poses still represent valid 3D poses as we just interpolate between the known frames. We determine the shape for each actor beforehand as we are aware of their height, weight and general body circumferences. We use a neutral body for every actor.

**SMPL Inpainting**: We utilize an unconditioned human motion diffusion model (MDM) [44], trained on AMASS [15], to inpaint the masked poses. For this, we subsample AMASS to 25Hz to match the frame-rate of our dataset. During inference of the diffusion model, we replace the unmasked keypoints by the annotated keypoints at each diffusion step. In this way, the annotated and verified 3D keypoints remain unchanged, but the occluded and masked parameters will be filled by the diffusion model.

Figure 3 provides an overview of our pose annotation process and we provide more details in the supplementary material.

### 3.1.3 Scene Annotation

We annotate the scene geometry either as 3D box or as cylinder, where we annotate trash bins, stools and circular tables as cylinders and everything else as box. In each of the four kitchens, we annotate the following 13 objects: **Whiteboard**, **Microwave**, **Kettle**, **Coffee Machine**, **Table** (sofa table, bar table, kitchen table), **Sittable** (sofas, arm chairs, chairs, and stools), **Cupboard** (floor and hanging cupboards), **Occluder** (walls and pillars), **Dishwasher**, **Drawer**, **Sink**, **Trash**, and **Out-of-Bound-Marker**, which marks the boundary of the visible area. The objects in the scene are annotated per frame since the objects can move during the recording. Fig. 2 shows four examples.

### 3.2 Statistics

Our dataset is a large-scale multi-person motion dataset, recorded at 25Hz, with over $4M$ individual 3D human poses of 90 individuals and over $650k$ frames, a total of 7.3h of recording, as summarized in Tab. 2. Compared to other datasets with 3D human poses, it contains more numbers of persons in the scene and more diverse activities recorded in a real environment as summarized in Tab. 1. Furthermore, the dataset not only includes the context of a static environment, but also moving objects that have been annotated. Each pose is further labeled with one or multiple activities out of 82 activity classes, for example, sitting in a chair, writing on a whiteboard, or washing hands[5]. Scene geometry is annotated per frame as either a 3D box or as 3D cylinder as well as with one out of 13 object classes, e.g., coffee machine, table, or whiteboard. The dataset was recorded in 4 kitchens, A, B, C and D, with common scene geometry such as coffee machines, chairs and whiteboards, but with

---

[4]`https://github.com/Dou-Yiming/Pose_to_SMPL`
[5]A full list is provided in the supplementary material

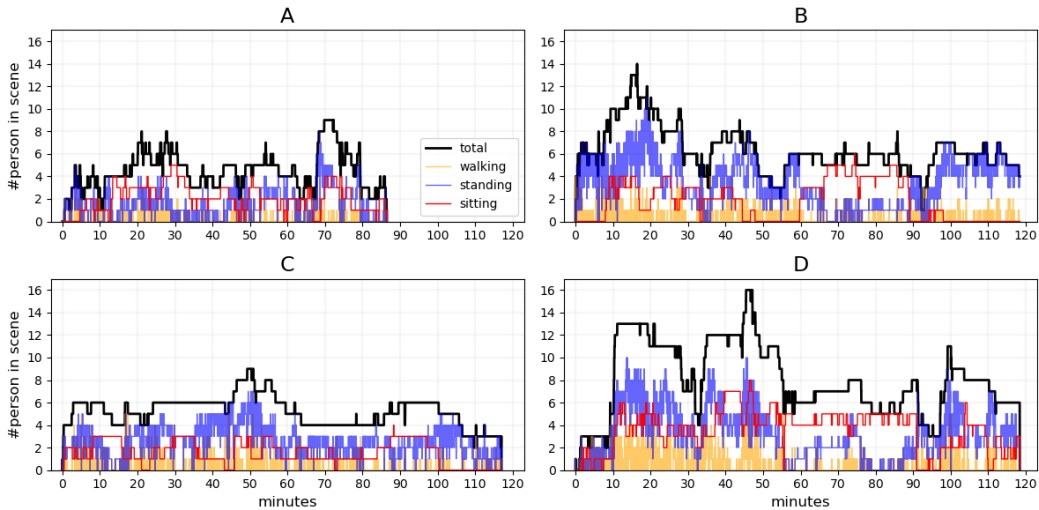

Figure 4: Number of persons that are *Walking*, *Standing* or *Sitting* at a frame. The x-axis represents the elapsed time in minutes while the y-axis represents the number of person that perform the corresponding activity. The black curve is the total number of persons at a frame.

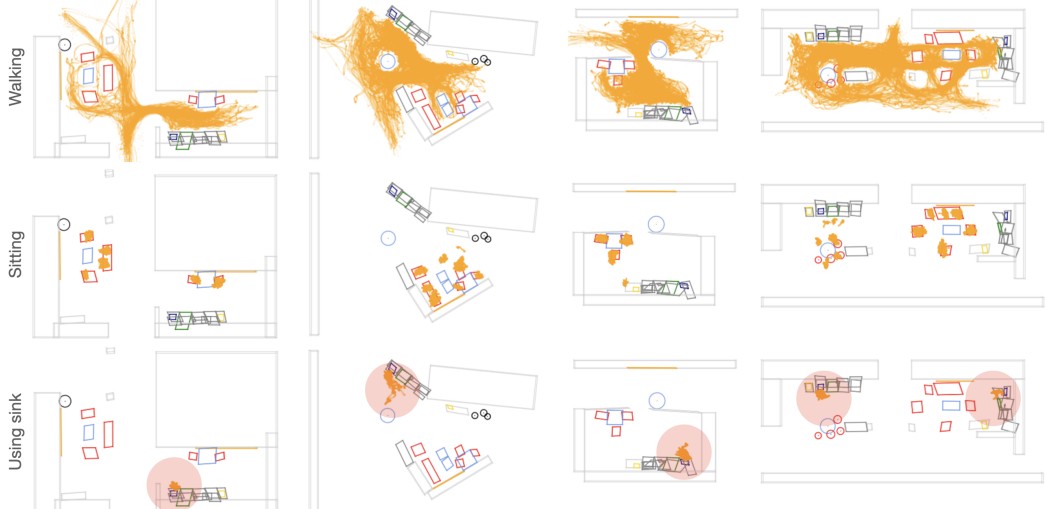

Figure 5: Occurrence map for the activities *Walking*, *Sitting* and *Using sink*. The maps are generated by plotting the location of the root joint of each person in the entire recording when persons perform the corresponding activity. For the last activity, *Using sink*, we highlight the location with a red circle as the activity only occurs close to the sink (dark blue).

different room layouts, as can be seen in Fig. 2. The number of annotated objects varies between 29 and 50. Each kitchen sequence has been recorded continuously, taking between $1.5$h (A) to $2h$ (B, C, D). While there are in average between $4.42$ and $7.97$ persons at the same time visible in a scene, the number of persons varies largely during a sequence since persons enter and leave the scene as shown in Fig. 4.

From the 82 annotated activities, we define 5 as posture activities: *Walking*, *Sitting*, *Standing*, *Leaning* and *Kneeling*. At each point in time, a person exhibits one of the 5 postures. We differentiate between *Leaning* and *Standing* by defining that a person leans if the person's weight is supported by a scene object, e.g., a cupboard. *Kneeling* is very rare and is only briefly observed in two kitchens: we decided to annotate it for completeness as it would not fit any of the four other postures. The postures greatly vary in frequency as we show in Tab. 3. The most common postures are *Sitting* and *Standing*, which is expected from natural social human interactions. In Fig. 4, we plot the three most common postures, *Walking*, *Standing* and *Sitting* over time. We observe that in B and D there is a time window at the end of the recording where most persons sit. This is not observed in A and C. This shows that

Table 3: Number of frames and persons per dataset for the posture activities *Walking*, *Sitting*, *Standing*, *Leaning* and *Kneeling*.

| Action | A | B | C | D | sum |
|---|---|---|---|---|---|
| *Walking* | 39,454 | 73,051 | 79,195 | 119,230 | 310,930 (3.45h) |
| *Sitting* | 265,521 | 314,791 | 263,652 | 631,673 | 1,475,637 (16.39h) |
| *Standing* | 209,525 | 635,835 | 464,940 | 569,712 | 1,880,012 (20.89h) |
| *Leaning* | 25,491 | 46,538 | 56,846 | 44,685 | 173,560 (1.9h) |
| *Kneeling* | 0 | 702 | 0 | 50 | 752 (30s) |

the distribution even of the basic posture activities varies over time and from scene to scene. In Fig. 5, we show a bird eye view for all four kitchens and plot where three example activities occur. As expected, *Walking* covers almost the entire scene while object-specific interactions with the scene such as *Sitting* and *Using sink* are localized at the corresponding scene objects. Although chairs move in the scene, they are not moved to completely different locations since the kitchens offer sufficient chairs.

### 3.3 License and Consent

The dataset and API are free to download[6] and can be used for non-commercial purposes. The API is under MIT-License while the dataset utilizes a custom license. All subjects signed forms consenting that recordings of them or derived of them can be used for non-commercial research purposes. The recording has been approved by the ethical review committee of the University of Bonn. In contrast to the raw data [41], Humans in Kitchens does not contain personally identifiable information. The raw data is also accessible[7], but requires to sign a license agreement and is subject to export regulations. Note that the raw data is not required for using Humans in Kitchens. The data does not contain any offensive content.

## 4 Experiments

We evaluate various state-of-the-art methods on the human motion forecasting task of our dataset. More precisely, the goal is to learn a function $f$ that takes as input a human pose sequence $\mathbf{X}^{1:t} = (\mathbf{x}_1^{1:t}, \mathbf{x}_2^{1:t}, \cdots, \mathbf{x}_n^{1:t})$ of $n$ persons, where $\mathbf{x}_i^{1:t} \in \mathbb{R}^{t \times (29 \times 3)}$ represents a 3D motion sequence of $t$ frames for person $i$, and forecasts the future motion for all $n$ persons in global coordinates:

$$\hat{\mathbf{X}}^{t+1:T} = f(\mathbf{X}^{1:t}), \quad \hat{\mathbf{X}}^{t+1:T} \in \mathbb{R}^{(T-t) \times n \times (29 \times 3)}. \tag{1}$$

As our dataset contains natural behavior over very long time, we select 2 interesting motion activities, namely *Walking* and *Sitting down*, and 4 interesting interactions with the objects *Whiteboard*, *Sink*, *Cupboard*, and *Coffee Machine* for evaluation. Note that most of these actions involve social interactions since the persons sit in groups and discuss at the whiteboard. For training, we use the kitchens A, B and C and we evaluate on D, which is the largest among the four kitchens. For the human-object interactions, we select the last observed frame $t$ as the first frame of the annotated action, while for the motion activities we select frame $t - 10$ such that a few frames of the motion are already observed. Overall, we sample all occurrences of the given activity in the test set[8]. We only evaluate the forecast motion of the person that performs the activity and not for any other person in the scene. This allows to report accuracy per activity, but it still requires to model the context of the other persons. We consider two distinct protocols: *short-term* and *long-term* motion forecasting. In the short-term protocol, we employ the widely used Mean Per Joint Positional Error (MPJPE) metric [45] to measure the positional disparity between the predicted motion and the ground truth. For the long-term protocol, we use the Normalized Directional Motion Similarity Score (NDMS) [46], which effectively assesses the quality of motion sequences of any given length.

**Baseline methods**: We evaluate 4 recent single-person human motion forecasting methods, namely siMLPe [47], CHICO [18], pgbig [48] and History-Repeats-Itself [49], and the multi-person forecasting method Multi-Range Transformers (MRT) [50]. While the single-person approaches forecast the

---

[6]https://github.com/jutanke/hik
[7]https://github.com/bonn-activity-maps/bonn_activity_maps
[8]More details are provided in the supplementary material

Table 4: MPJPE ↓ in dm. Methods denoted with * forecast all persons at the same time.

| frame | 5 | 15 | 25 | 5 | 15 | 25 | 5 | 15 | 25 | 5 | 15 | 25 | 5 | 15 | 25 | 5 | 15 | 25 |
|---|---|---|---|---|---|---|---|---|---|---|---|---|---|---|---|---|---|---|
| | walking | | | sitting down | | | whiteboard | | | sink | | | cupboard | | | coffee | | |
| MRT* [50] | 0.40 | 0.91 | **1.40** | 0.38 | 0.97 | 1.42 | 0.24 | 0.56 | 0.80 | 0.29 | 0.80 | 1.16 | 0.28 | 0.61 | **0.89** | 0.26 | 0.69 | 1.19 |
| siMLPe [47] | 0.39 | 0.97 | 1.59 | 0.37 | 0.98 | 1.40 | 0.24 | 0.53 | 0.72 | 0.17 | 0.43 | 0.59 | 0.23 | 0.64 | 0.98 | 0.22 | 0.55 | 0.88 |
| CHICO [18] | 0.37 | 0.88 | 1.42 | 0.36 | 0.96 | 1.36 | 0.27 | 0.55 | 0.69 | 0.19 | 0.46 | 0.61 | 0.23 | 0.65 | 1.05 | 0.22 | 0.55 | **0.86** |
| pgbig [48] | 0.34 | 0.85 | **1.40** | 0.34 | 0.87 | **1.22** | 0.23 | 0.52 | **0.66** | 0.16 | 0.42 | 0.58 | 0.20 | 0.59 | 0.93 | 0.20 | 0.54 | **0.86** |
| HistRep [49] | **0.30** | **0.83** | 1.57 | **0.30** | **0.84** | 1.23 | **0.19** | **0.48** | 0.68 | **0.12** | **0.40** | **0.57** | **0.16** | **0.57** | 0.99 | **0.17** | **0.50** | 0.95 |

Table 5: NDMS ↑. Methods denoted with * forecast all persons at the same time.

| frame | 1 | 25 | 100 | 200 | 250 | 1 | 25 | 100 | 200 | 250 | 1 | 25 | 100 | 200 | 250 |
|---|---|---|---|---|---|---|---|---|---|---|---|---|---|---|---|
| | walking | | | | | sitting down | | | | | whiteboard | | | | |
| MRT* [50] | 0.81 | 0.34 | 0.18 | 0.13 | 0.12 | 0.81 | 0.28 | 0.14 | 0.11 | 0.10 | 0.80 | 0.26 | 0.14 | 0.11 | 0.10 |
| siMLPe [47] | **0.87** | 0.41 | 0.24 | 0.20 | 0.19 | 0.87 | 0.37 | 0.22 | 0.19 | 0.18 | 0.88 | 0.35 | 0.21 | 0.18 | 0.17 |
| CHICO [18] | 0.84 | 0.40 | **0.25** | 0.20 | 0.19 | 0.85 | 0.36 | 0.22 | 0.19 | 0.18 | 0.86 | 0.35 | 0.22 | 0.18 | 0.17 |
| pgbig [48] | 0.86 | 0.43 | **0.25** | **0.21** | **0.20** | 0.87 | 0.38 | **0.23** | **0.20** | **0.19** | 0.87 | 0.37 | **0.23** | **0.20** | **0.19** |
| HistRep [49] | **0.87** | **0.46** | 0.21 | 0.15 | 0.14 | **0.88** | **0.43** | 0.19 | 0.14 | 0.13 | **0.89** | **0.43** | 0.19 | 0.14 | 0.13 |
| | sink | | | | | cupboard | | | | | coffee | | | | |
| MRT* [50] | 0.81 | 0.26 | 0.13 | 0.10 | 0.10 | 0.81 | 0.29 | 0.15 | 0.11 | 0.10 | 0.82 | 0.30 | 0.16 | 0.12 | 0.11 |
| siMLPe [47] | **0.90** | 0.36 | 0.22 | 0.19 | 0.18 | 0.86 | 0.37 | 0.22 | 0.19 | 0.18 | **0.86** | 0.38 | 0.22 | 0.19 | 0.19 |
| CHICO [18] | 0.87 | 0.34 | 0.22 | 0.19 | 0.18 | 0.84 | 0.36 | 0.22 | 0.19 | 0.18 | 0.83 | 0.36 | 0.23 | 0.19 | 0.18 |
| pgbig [48] | 0.89 | 0.37 | **0.23** | **0.20** | **0.20** | 0.86 | 0.39 | **0.23** | **0.20** | **0.19** | 0.85 | 0.39 | **0.24** | **0.21** | **0.20** |
| HistRep [49] | **0.90** | **0.43** | 0.19 | 0.14 | 0.13 | **0.87** | **0.43** | 0.19 | 0.14 | 0.13 | **0.86** | **0.44** | 0.20 | 0.15 | 0.14 |

motion of each person independently, MRT forecasts the motion of all persons jointly. The source code of all methods is publicly available and we utilize the original hyper-parameters. We only adjust the input and output dimensions to fit our skeleton representation. The adapted source code of all methods is publicly accessible via our API. We normalize the human poses in a pre-processing step as in [46] such that the root joint of the last observed frame $t$ is translated to the origin and the hip is axis-aligned with the x-axis. After the forecasting, we apply the inverse transformation to convert the forecast motion back into global 3D coordinates. For MRT, we use the normalization proposed in [50], i.e., we shift all kitchens so that the mean pose location is at the origin to prevent drift.

**Short-Term Forecasting**: For short-term forecasting, the methods receive 50 frames (2s) of input motion and forecast 25 frames (1s). As metric we utilize MPJPE. The results in Tab. 4 show that History-Repeats-Itself [49] performs best for the first 15 frames, but at frame 25 pgbig [48] performs best for most activities. MRT [50] performed best on the *Cupboard* sequences at frame 25. We will see in the long-term experiments that History-Repeats-Itself is very strong for very short time horizons but not suitable for longer sequences, whereas pgbig [48] is a more general approach that performs well also for longer horizons. In general, the lowest errors are observed for *Whiteboard*, *Sink*, *Cupboard* and *Coffee* since the global position changes less than for the activities *Walking* and *Sitting down*.

**Long-Term Forecasting**: For long-term forecasting, the methods forecast 250 frames (10s) given 50 frames (2s). In general, we suggest for future works to use 250 frames for the observation as well since restricting the observations to 50 frames is not necessary. However, state-of-the-art approaches operate on a fixed input window, an important hyper-parameter, that sometimes can only be changed by modifying the architecture. We thus kept the input sequence as for the short-term forecasting. As metric, we utilize NDMS with kernel size 8 [46]. The results in Tab. 5 show that all methods produce a reasonable motion up to 1 second but the quality deteriorates afterwards. Similar to the short-term forecasting, History-Repeats-Itself [49] performs best for the shorter time horizons while pgbig [48] performs best for the longer time horizons. Interestingly, MRT [50], which is the only multi-person method, performs worse than the other methods. The difference might be the normalization used in MRT, which is based on the mean pose and not the last observed pose. In general, MRT generates good motions for dynamic activities, i.e., walking, sitting down, or standing up, but for motions, which are more subtle, the model often freezes. This suggests that the joint generation of realistic motion of more than 3 persons is challenging. We also find that MRT struggles to keep sensible distances between persons when a scene gets more crowded. For a high number of persons per scene, i.e., more than 8, we found multiple instances of a person walking into another person. This does not occur in scenes with fewer persons. This indicates that MRT is able to utilize information about the distances and relative locations between people, but its performance of doing so deteriorates with more people. Overall, the results show that modeling the interactions of multiple persons is not well handled by the current state-of-the-art. Furthermore, none of the current approaches is able to incorporate scene context. Humans in Kitchens thus opens new research directions to study multi-person human motion forecasting with scene context.

## 4.1 Discussion

We presented Humans in Kitchen, a large-scale dataset for multi-person human motion forecasting, consisting of 7.3h hours of recordings of persons interacting in real kitchen environments. The dataset includes over 4M annotated 3D human poses of 90 individuals with unique IDs, 82 annotated activities and 156 annotated objects. It is the first dataset that combines multiple persons, activities, and scene contexts in a real setting. We utilize well-known open source formats and the dataset can be easily used with our provided API. We further provided two benchmarks for short and long-term human motion forecasting. We belief that Humans in Kitchens will be a valuable source to advance and evaluate approaches for multi-person human motion forecasting with scene context.

**Limitations and Future Directions**: While Humans in Kitchens aims to record behaviour as natural as possible, persons might still behave differently from real-life when they know that they are recorded. Due to the more natural behaviour, the majority of activities is standing and sitting as shown in Tab. 3. Such unbalanced data is realistic and it can be addressed by evaluating per activity to avoid a bias towards these two activities. Another limitation is the coarse scene geometry annotation. We decided to prefer a coarse but dynamic geometry compared to a static geometry since in particular chairs are moved within a 2 hours recording. It might be possible to fit 3D shapes to the objects in the future, but this will not be straightforward since the objects rotate and the orientation of the kettle, for example, is difficult to annotate. Furthermore, we only fit the SMPL model and do not use more detailed body models that include hand pose or more details of the face due to the large degree of occlusion as well as the limited resolution caused by the distance to the cameras. The dataset can thus not be used for modeling fine-grained hand-object interactions. We have also observed that state-of-the-art approaches struggle to model the motion and interaction of multiple persons. Another open research question is the modeling of scene context for forecasting, in particular as the size of objects varies from 30cm (kettle) to 10m (walls) in width.

**Negative Societal Impact**: The purpose of this dataset is to aid research for forecasting approaches for socially aware robots. It needs to be considered that forecasting models can also be misused for monitoring civilians. To address this issue, we focused only on kitchen related activities. In case of misuse, we reserve our right to withdraw the permission to use the dataset at any point.

## Acknowledgments and Disclosure of Funding

The work has been funded by the Deutsche Forschungsgemeinschaft (DFG, German Research Foundation) 1927/5-2 (FOR 2535 Anticipating Human Behavior) and the ERC Consolidator Grant FORHUE (101044724). Oh-Hun Kwon has been supported by the Deutsche Forschungsgemeinschaft (DFG, German Research Foundation) under Germany's Excellence Strategy - EXC 2070 -390732324. The development of this publication was supported by the Ministry of Economic Affairs, Industry, Climate Action and Energy of the State of North Rhine-Westphalia as part of the flagship project ZERTIFIZIERTE KI.

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
