# Supplementary Materials: Humans in Kitchens: A Dataset for Multi-Person Human Motion Forecasting with Scene Context

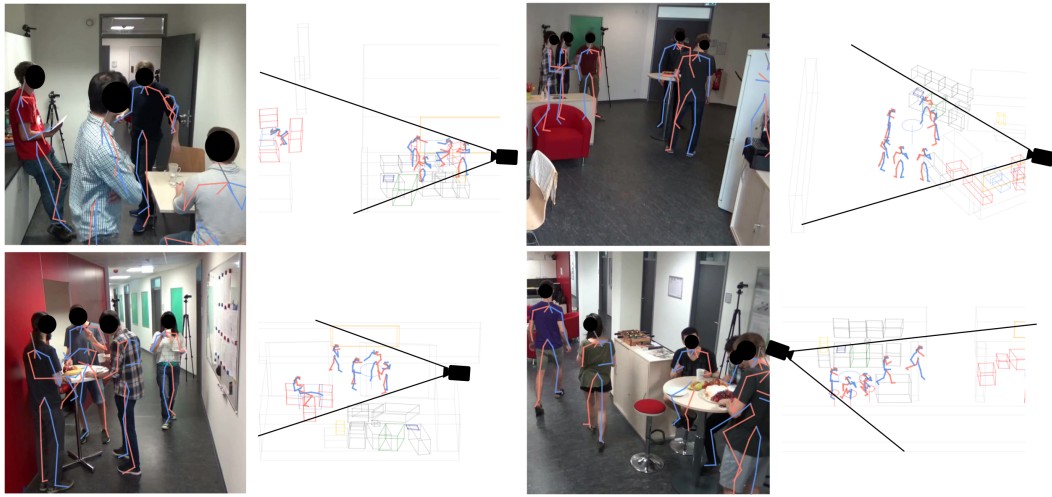

Figure 1: Sample scenes with 3d human poses projected onto camera views for each kitchen.

# 1 Dataset

## 1.1 Dataset Structure

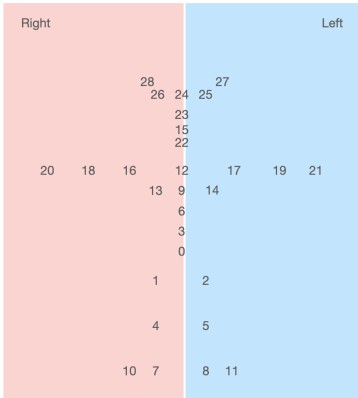

Figure 2: 3d skeleton with 29 joints. The first 24 joints are equivalent to the SMPL pose [1] while joints 25 to 28 represent the head, with 24 being the nose, 25 and 26 being the eyes and 27 and 28 being the ears, similar to the OpenPose skeleton.

37th Conference on Neural Information Processing Systems (NeurIPS 2023) Track on Datasets and Benchmarks.

Our dataset contains two folders, `poses` and `scenes`. The `poses` subfolder contains `npz` files of the form `{dataset}_{person identification number (pid)}_{sequence number}.npz` where `dataset` determines one of the kitchens A, B, C or D, `pid` determines a unique identifier number per dataset for a given individual and where `sequence number` is a simple counter. Each file represents the entire motion from when a person enters the scene until they leave. As persons may enter and leave multiple times a new file is created whenever they re-enter the scene. The person can be uniquely identified by their `pid` while the `sequence number` tracks the amount of times they have re-entered the scene. Each file contains the following tensors:

- `betas`: 10; represent the SMPL [1] shape of this person
- `smpl`: $t \times 21 \times 3$; represent the SMPL [1] pose in axis-aligned angular representation
- `transforms`: $t \times 6$; represents global rotation and translation of the SMPL pose with the global rotation as rotation vector
- `poses3d`: $t \times 29 \times 3$; 3d joint locations of each pose in global 3d space - this is equivalent of applying forward kinematics with global transform, using the entries from `smpl` and `transforms`. A sample skeleton can be seen in Figure 2.
- `frames`: $t$; frame number in actual dataset time
- `act`: $t \times 82$; action annotations, where 1 determines an action and 0 its absence.

where $t$ determines the total amount of frames of that sequence.

The `scenes` folder contains a subfolder for each of the four kitchens A, B, C, and D, which has two files, `{scene object name}.json` and `{scene object name}.npy` for each scene object where `scene object name` acts as unique identifier. The `json` file contains meta information such as which of the 13 object classes the object is and what shape, box or cylinder, the object is approximated by. The `npy` file contains the location(s) of the object for each frame in the dataset. For box shaped objects, the data is represented as a 3d box while for cylinders the object is determined by a 3d point and a radius where the 3d point defines the center top point of the cylinder. We define the floor to be at $z = 0$ and all cylinder objects stand on the floor (trash bins and barstools), while boxes can be above the floor, e.g., the coffee machine standing on a cupboard or a cupboard on the wall.

The python API[1] is the recommended way of using Humans in Kitchens.

**Human Body Representation**: Skinned Multi-Person Linear Model (SMPL) [1] have emerged as de facto standard to represent parameterized human bodies. SMPL has enjoyed increased popularity in 3d human motion modeling [2] in recent years thanks to the large-scale AMASS [3] dataset. Utilizing a parametric body model over 3d joint locations has two advantages for 3d human motion modeling: it provides a separation between local pose, body shape and global transformation, making reasoning about a pose independent of its location in 3d space. On top of that, SMPL's shape parameter determines limb length ensuring that the body skeleton remains consistent across time. This is especially relevant for preventing limb length drift when the scene spans a large area with varying camera cover and persons are in the scene for a long time. For this reason, we employ SMPL as body representation.

## 1.2 Responsibility

We bear all responsibility in case of violation of rights. We license our dataset under a custom license which allows subjects to withdraw their agreement, triggering a deletion of the video sequences in the dataset. In case of a withdrawal, we will make a notification on the dataset website and contact all researchers via email who have downloaded the dataset. For this reason, the users of the dataset need to provide their email addresses. Please note that the dataset can be used without the video data. A withdrawal of the skeleton data is very unlikely. Nevertheless, we have implemented procedures in case of a withdrawal. Our code is open-sourced under the MIT license.

## 1.3 Dataset License

```
The dataset can only be used for non-commercial use such as teaching,
academic research, public demonstrations and personal experimentation.  You
```

---

[1] `https://github.com/jutanke/hik`

may not use this dataset or any derivative works in any form for commercial purposes.  Examples of commercial purposes would be running business operations, licensing, leasing, or selling the dataset, distributing the dataset for use with commercial products, using the dataset in the creation or use of commercial products or any other activity which purpose is to procure a commercial gain to you or others.  The dataset shall not be copied, shared, distributed, re-sold, offered for re-sale, transferred or sub-licensed in whole or in part except that you may make one copy for archive purposes only.  The subjects of the recording have the right to withdraw the permission to use the dataset at any time.  In this case, the corresponding data needs to be deleted within 14 days after receiving such a request.  This also includes any copies for archive purposes.  Images or videos of the dataset can be used and modified for scientific publications, supplementary material for scientific publications, presentations, and public demonstrations.

### 1.4   Hosting and Maintenance Plan

The dataset is hosted as downloadable link on Sciebo[2], a cloud storage for universities and higher education institutes that is funded by the state NRW and ensures long-term storage. Copies of the dataset are further archived on the servers of the Institute of Computer Science. The API is available on GitHub[3] and users are encouraged to utilize the GitHub *issues* to ask questions.

## 2   Camera Calibration

Calibrating the intrinsic parameters for each camera is a straightforward process using a checkerboard and the algorithm described by Zhang [4]. However, estimating the extrinsic parameters for multiple cameras positioned arbitrarily to cover a large and non-convex area presents a challenge due to the lack of structured targets visible to all cameras.

To address this challenge, we adopt a different approach. Instead of designing complex calibration targets or tracking moving objects, we utilize the existing rigid landmarks in the scene. These landmarks, such as corners of cupboards, whiteboards, and fire warning signs, serve as reference points in the 3d space. Assuming that all cameras remain static during the capture process, we can easily annotate these landmarks on the 2d frames of each camera view.

To determine the 3d location of these landmarks in a joint global coordinate space, we employ multilateration techniques [5]. Using an off-the-shelf laser measuring device from several known points, we obtain distances from known positions to the landmarks. To ensure accurate distance measurement, the laser measuring device is securely attached to a gimbal with an omni-directional rotational joint. By solving the kinematics of the gimbal, we can determine the precise distance between the target landmark and the rotation center of the gimbal. With the distances obtained from at least three known positions, we can accurately determine the positions of the landmarks in 3d space by optimizing an error function

$$L(x, y, z) = \frac{1}{N} \sum_{i}^{N} \left\| \sqrt{(x - x_i)^2 + (y - y_i)^2 + (z - z_i)^2} - r_i \right\|_2^2 \tag{1}$$

where $x_i$, $y_i$, and $z_i$ represent the base points, and $r_i$ is the measured distance. This error function minimizes the difference between the measured distances $r_i$ and the Euclidean distances between the landmark positions $(x, y, z)$ and the known positions $(x_i, y_i, z_i)$. By optimizing this error function (1) with gradient descent, we can accurately determine the positions of the landmarks in the 3d space.

The process of estimating the landmark points in 3d space consists of the following steps. First, we measure a set of base points $\hat{X}_i$ on the ground plane ($z = 0$) as an initial set of landmarks. These base points should be selected in such a way that the laser measuring device can reach at least three of the estimated points on the ground plane to the target landmarks in 3d space. To establish the

---

[2]https://hochschulcloud.nrw/
[3]https://github.com/jutanke/hik

ground plane, we choose a point on the ground as the origin and select one point on the $x$-axis and another on the $y$-axis of our right-handed coordinate frame. By considering distances from more than two ground points, the landmarks on the ground plane can be located by optimizing the error function (1) while imposing the constraints $z = 0$ and $z_i = 0$.

Next, we measure all the target landmarks in 3d space from the base points on the ground. It is important to note that, in this step, we use the laser measuring device attached to the gimbal to ensure that the measured distance is between the target point and the rotational center of the gimbal. Using the measured distances, we optimize the error function (1) in a similar manner.

By performing these steps, we can accurately estimate the positions of the landmarks in 3d space and then utilize the annotated 2d points and the estimated 3d points to solve the Perspective-$n$-Point (P$n$P) problem [6], which allows us to estimate the extrinsic parameters of each camera.

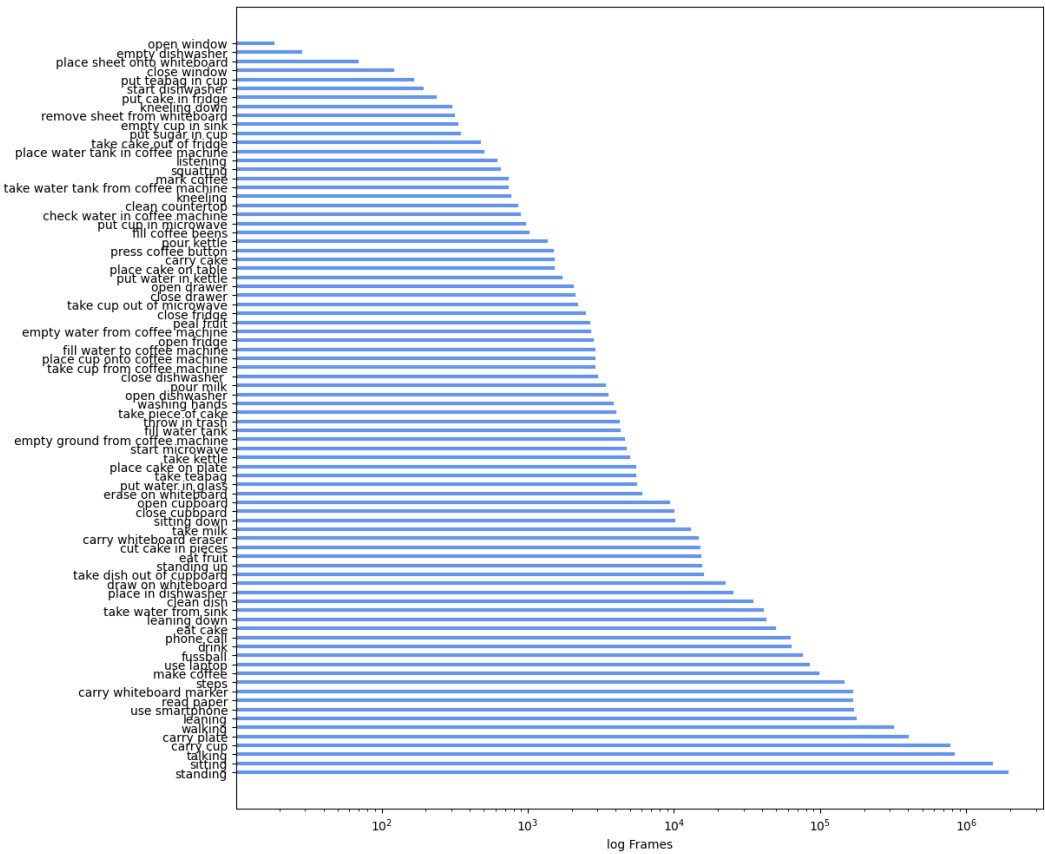

Figure 3: Number of frames in the dataset for all 82 activites, the x-axis represents the number of frames on a log-scale.

## 3   Activities

Below, we list all 82 annotated activities and their annotation instructions. While Figure 3 shows how often each action occurs, Figures 4 and 5 show additional activity occurrence maps.

1. **carry cake**: starts when person touches the cake plate and ends when the person releases the plate
2. **carry cup**: starts when person hand touches and ends when person releases it
3. **carry plate**: starts when person hand touches and ends when person releases it
4. **carry whiteboard eraser**: starts when person hand touches and ends when person releases it

5. **carry whiteboard marker**: starts when person hand touches and ends when person releases it

6. **check water in coffee machine**: starts when person touches water tank and ends once the full water tank is returned to the coffee machine

7. **clean countertop**: starts when person hand touches countertop and ends when person is done cleaning and lifts the hand from countertop

8. **clean dish**: starts when person hand touches and ends when person releases plate

9. **close cupboard**: starts when person touches cupboard door and ends when cupboard is closed

10. **close dishwasher**: starts when person touches dishwasher door and ends when dishwasher is closed

11. **close drawer**: starts when person touches drawer and ends when drawer is closed

12. **close fridge**: starts when person touches fridge and ends when fridge is closed

13. **close window**: starts when person touches window and ends when window is closed

14. **cut cake in pieces**: starts when knife touches cake and ends when the knife is lifted off the cake

15. **draw on whiteboard**: starts when marker touches whiteboard and ends when marker is lifted off whiteboard

16. **drink**: starts when glass touches mouth and ends when glass is lifted off mouth

17. **eat cake**: starts when fork touches mouth and ends when fork is taken out of mouth

18. **eat fruit**: starts when fruit is being placed in mouth and ends when hand releases the fruit

19. **empty cup in sink**: starts when cup is turned and ends when cup is rotated up again

20. **empty dishwasher**: starts when dishwasher is opened and ends when dishwasher is closed

21. **empty ground from coffee machine**: starts when ground tray is taken out of machine and ends when tray placed back

22. **empty water from coffee machine**: starts when water is being poured out of coffee machine water box and ends when box is stopped being poured.

23. **erase on whiteboard**: starts when eraser touches whiteboard and ends when eraser is lifted off whiteboard

24. **fill coffee beans**: starts when person takes coffee beans box out of coffee machine and ends when box is placed back into machine

25. **fill water tank**: starts when the coffee machine water box is placed in the sink and the faucet is opened. Ends when the faucet is closed.

26. **fill water to coffee machine**: starts when water box is taken out of coffee machine and ends when placed back

27. **fussball**: starts when person touches the table football, ends when person releases table football

28. **kneeling**: starts when person is kneeling, ends when person initiates standing up

29. **kneeling down**: starts when person initiates kneeling down, ends when person is kneeling

30. **leaning**: starts when person is leaning to object, e.g. cupboard, ends when person initiates standing up

31. **leaning down**: starts when leaning down is initiated, ends when person is leaning (on an object)

32. **listening**: starts when another person is talking and this person looks at them, ends when other person either stops talking or when this person looks away.

33. **make coffee**: starts when person places coffee mug in coffee machine, ends when coffee mug is taken out

34. **mark coffee**: starts when person "pays" their coffee by marking on a sheet, starts when pen touches paper, ends when pen is released

35. **open cupboard**: starts when hand touches cupboard, ends when cupboard fully opened

36. **open dishwasher**: starts when hand touches dishwasher, ends when dishwasher fully opened

37. **open drawer**: starts when hand touches drawer, ends when drawer fully opened

38. **open fridge**: starts when hand touches fridge, ends when drawer fridge opened

39. **open window**: starts when hand touches window, ends when drawer window opened
40. **peal fruit**: starts when person starts pealing, ends when pealing is done
41. **phone call**: starts when person takes phone to ear, ends when person takes phone off ear
42. **place cake on plate**: starts when pastry fork touches cake piece, ends when cake piece is placed onto plate
43. **place cake on table**: starts when cake is carried to table from fridge, ends when cake is released
44. **place cup onto coffee machine**: starts when mug touches coffee machine, ends when hand is released
45. **place in dishwasher**: starts when dishwasher is opened, ends when dishwasher is closed
46. **place sheet onto whiteboard**: starts when hand/paper touches whiteboard, ends when hand is lifted off whiteboard
47. **place water tank in coffee machine**: starts when water tank is lifted over coffee machine, ends when hand is released from tank
48. **pour kettle**: starts when water starts flowing, ends when water stops flowing
49. **pour milk**: starts when milk starts flowing, ends when milk stops flowing
50. **press coffee button**: starts when finger touches button, ends when finger is released
51. **put cake in fridge**: starts when hand touch cake on table, ends when fridge door is closed
52. **put cup in microwave**: starts when cup is placed in mircowave, ends when cup is taken out
53. **put sugar in cup**: starts when pouring, ends when pouring ends
54. **put teabag in cup**: starts when teabag touches the water, ends when hand releases it
55. **put water in glass**: starts when faucet is on, ends when faucet turned off
56. **put water in kettle**: starts when faucet is on, ends when faucet turned off
57. **read paper**: starts when person looks at paper, ends when person looks away
58. **remove sheet from whiteboard**: starts when person touches sheet, ends when sheet is not touching whiteboard
59. **sitting**: starts when person is firmly sitting on the chair, ends when person initiates standing up
60. **sitting down**: starts when person initiates sitting down, ends when person is sitting
61. **squatting**: starts when person is squatting, ends when person is initiating standing up
62. **standing**: starts when person transitions from another pose (walking, sitting, etc) to standing, and after "standing up". Ends when another pose motion is initiated
63. **standing up**: starts when a person initiates a standing up-motion, ends when person stands
64. **start dishwasher**: starts when dishwasher is closed, ends when person steps away from dishwasher
65. **start microwave**: starts when person presses microwave button, ends when person hand lifted off button
66. **steps**: starts when person moves foot in direction, ends when person is standing. This is different from walking in that walking contains multiple steps, and is directed.
67. **take cake out of fridge**: starts when fridge is opened, ends when fridge closed
68. **take cup from coffee machine**: starts when hand touches cup and ends when cup is lifted from coffee machine
69. **take cup out of microwave**: starts when microwave door is opened and ends when cup leaves the microwave
70. **take dish out of cupboard**: starts when cupboard door is opened and ends as soon as dish is outside the cupboard.
71. **take kettle**: starts when person touches kettle and ends when person releases kettle
72. **take milk**: starts when person touches milk and ends when person releases milk
73. **take piece of cake**: starts when person holds a plate with cake on it and ends when person releases the plate
74. **take teabag**: starts when person grabs teabag and ends when teabag is dropped
75. **take water from sink**: starts when faucet is on, ends when faucet turned off

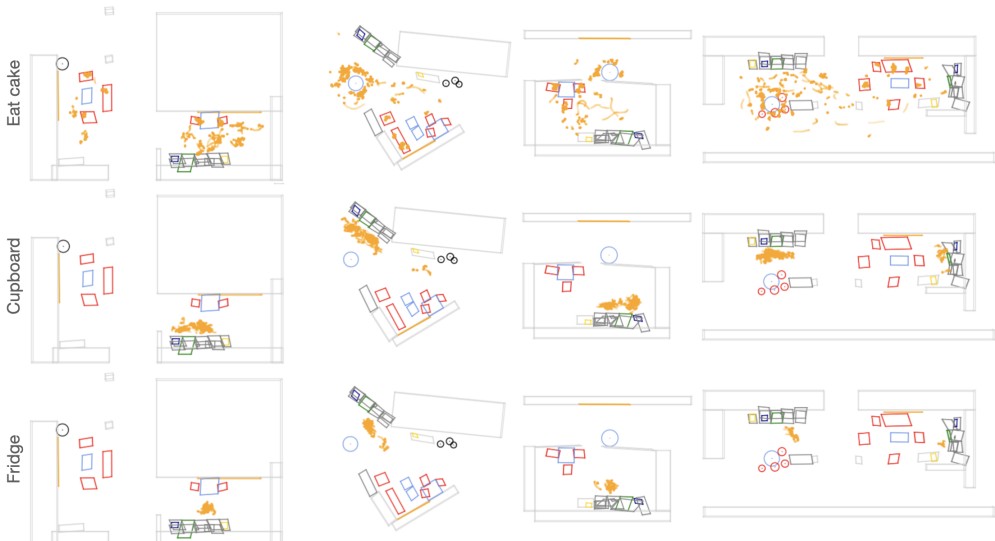

Figure 4: Activity occurrence maps for *Eat cake*, *Cupboard* (open/close cupboard), and *Fridge* (open/close fridge).

76. **take water tank from coffee machine**: starts when person lifts water tank, ends when person puts back water tank into coffee machine
77. **talking**: starts when person talks, ends when person stops talking
78. **throw in trash**: starts the frame the person drops the item and ends once the item hits the trash bin.
79. **use laptop**: starts when person has hands on laptop and looks at it, ends when person stands up or looks away.
80. **use smartphone**: starts when person looks on phone or phone calls, ends when person puts phone away
81. **walking**: starts when walking begins, ends when the last moving leg stops moving
82. **washing hands**: starts when hand under running water, ends when hands out of running water

In 27% of all frames we have activities that require at least two persons, like talking or writing on a whiteboard (subjects were asked to explain something to others). Furthermore, many activities are often induced by the actions of others, such as cutting cake or standing up from a chair. Figure 7 shows an example where two persons are engaged in a conversation while also drinking coffee and eating fruits. Figure 8 shows 3 actors being engaged by one person writing to the whiteboard.

# 4 Experiments Details

For our evaluation protocol, we select the 6 interesting actions walking, sitting down, whiteboard, sink, cupboard and coffee, as described in the paper. We report the average MPJPE and NDMS over all sequences of a given action class as the number of frames where an action is happening varies greatly. As several of our activity annotations represent interactions with the same object (e.g. opening and closing a cupboard, writing or erasing on whiteboard), we define the object interactions by grouping several activities:

1. **Walking**: 581 sequences
2. **Sitting down**: 99 sequences
3. **whiteboard**: 89 sequences, activities: "draw on whiteboard", "erase on whiteboard"
4. **sink**: 36 sequences, activities: "clean dish", "empty cup in sink", "put water in glass", "put water in kettle", "washing hands"
5. **cupboard**: 133 sequences, activities: "open drawer", "open cupboard", "close cupboard", "close drawer"

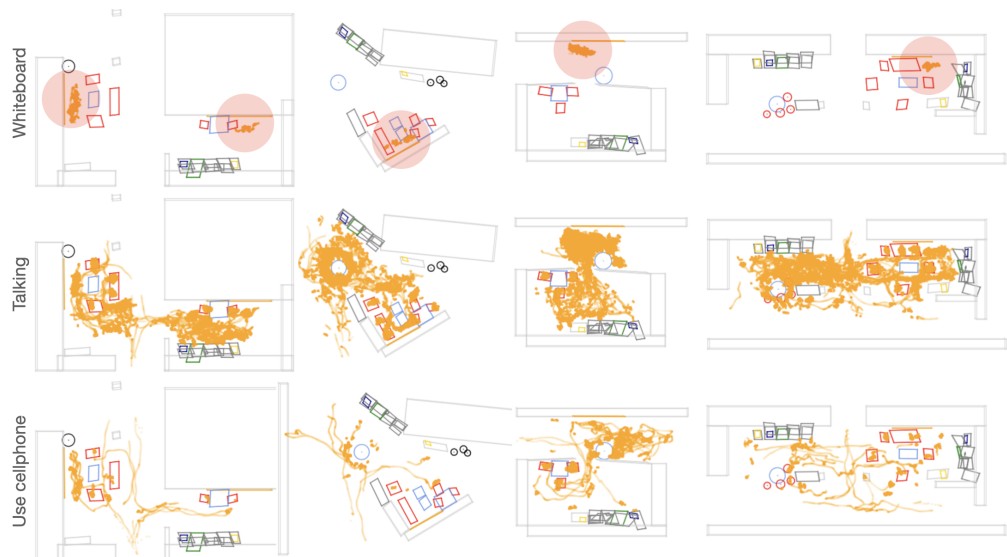

Figure 5: Activity occurrence maps for *Whiteboard* (write/erase whiteboard), *Talking*, and *Use cellphone*.

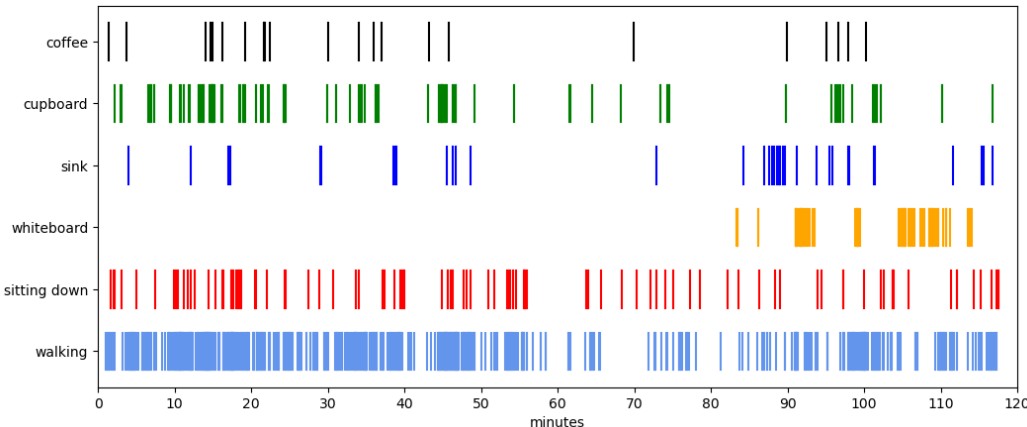

Figure 6: Occurrence of the evaluation classes throughout the recording in kitchen D. The x-axis is the elapsed time in minutes while the y-axis marks the evaluation class. Each line marks an occurrence.

6. **coffee**: 23 sequences, activities: "make coffee", "place cup onto coffee machine", "press coffee button", "take cup from coffee machine"

In Figure 6, we show when the 6 evaluation classes occur in our evaluation sequence. We observe that all evaluation classes except for whiteboard are evenly spread throughout the sequence, while the whiteboard was used only at the end of the recording session.

## 5    Annotation Details

For annotation, we utilize three custom-made annotation tools: a **3d nose annotation tool** (see Figure 7) where the noses of subjects are annotated per frame, an **activity annotation tool** (see Figure 8) where the 82 activities are annotated per frame per person, and a **3d pose annotation tool** to correct missing, incomplete, or erroneous 3d human poses. We also utilize the 3d pose annotation tool to annotate 3d boxes and cylinders. This can be easily achieved by casting 3d boxes and cylinders as their own *poses* where a 3d box is defined by 3 points and a cylinder by two, the center and a point on the perimeter. For cylinders, we always assume that they start on the floor ($z = 0$).

The instructions for the activity annotations are listed in Section 3. Annotators where not asked to annotate all 82 activities at the same time but only a single or two activities to prevent fatigue.

Two activities were annotated with highly correlated activities such as *Open cupboard* and *Close cupboard* or *Write to whiteboard* and *Erase from whiteboard*. Posture activities *Walking*, *Sitting*, *Standing*, *Leaning* and *Kneeling* as well as their intermediate activities (*Standing up*, *Sitting down*) where annotated per person all at once.

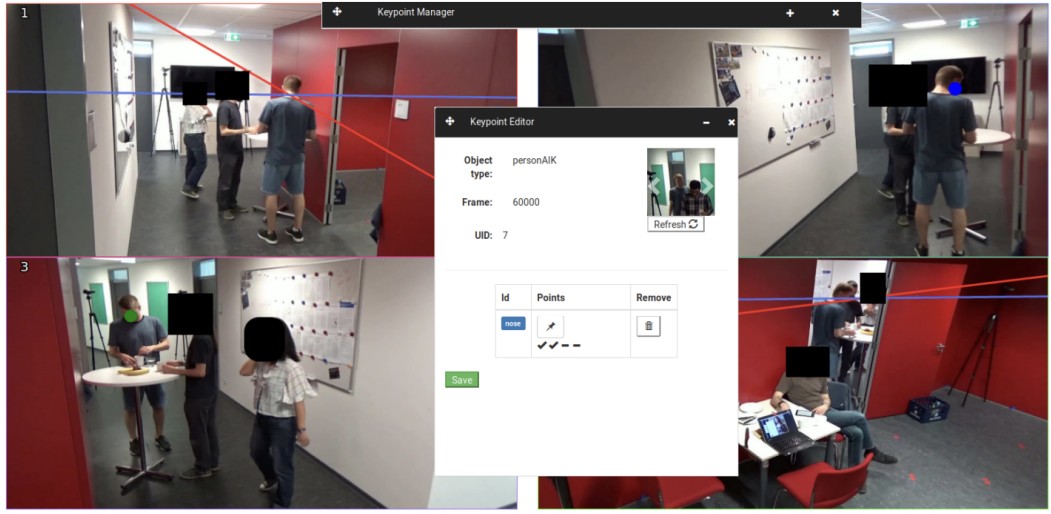

Figure 7: **3d nose annotation tool**: First, all persons have to be uniquely identified in our dataset. For this, we annotate for all individuals in the scene and for every frame their nose utilizing a self-developed 3d annotation tool. In this tool, annotators can choose up to 4 cameras, and then select the nose of the person with a mouse click. The epipolar lines of each annotated camera are projected onto the other camera views.

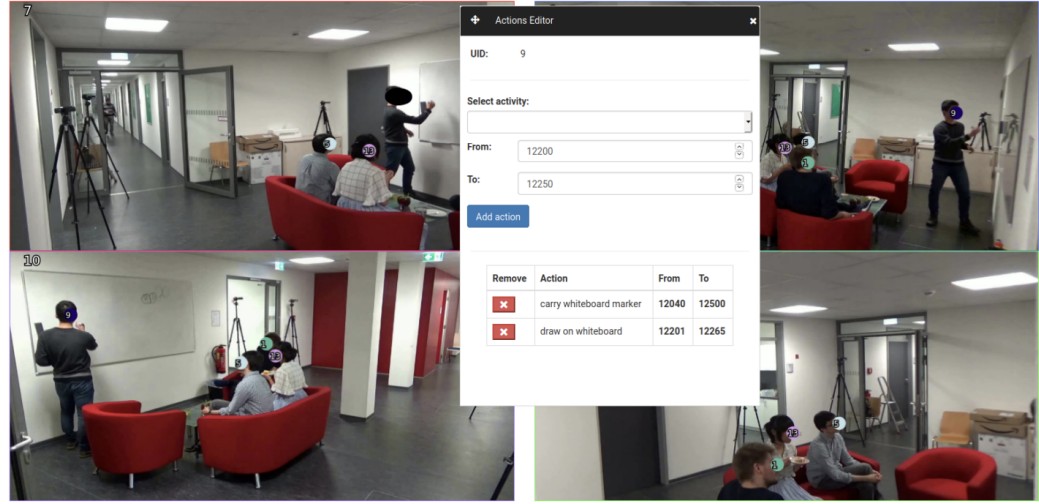

Figure 8: **Activity annotation tool**: To annotate per-frame activities for an individual, annotators can select up to 4 camera views where the activity can be selected from a drop-down menu and where a start and end frame can be set. Annotators can skim through the data using video player features such as *play*, *pause*, *drag frame selection* and *jump to frame*.

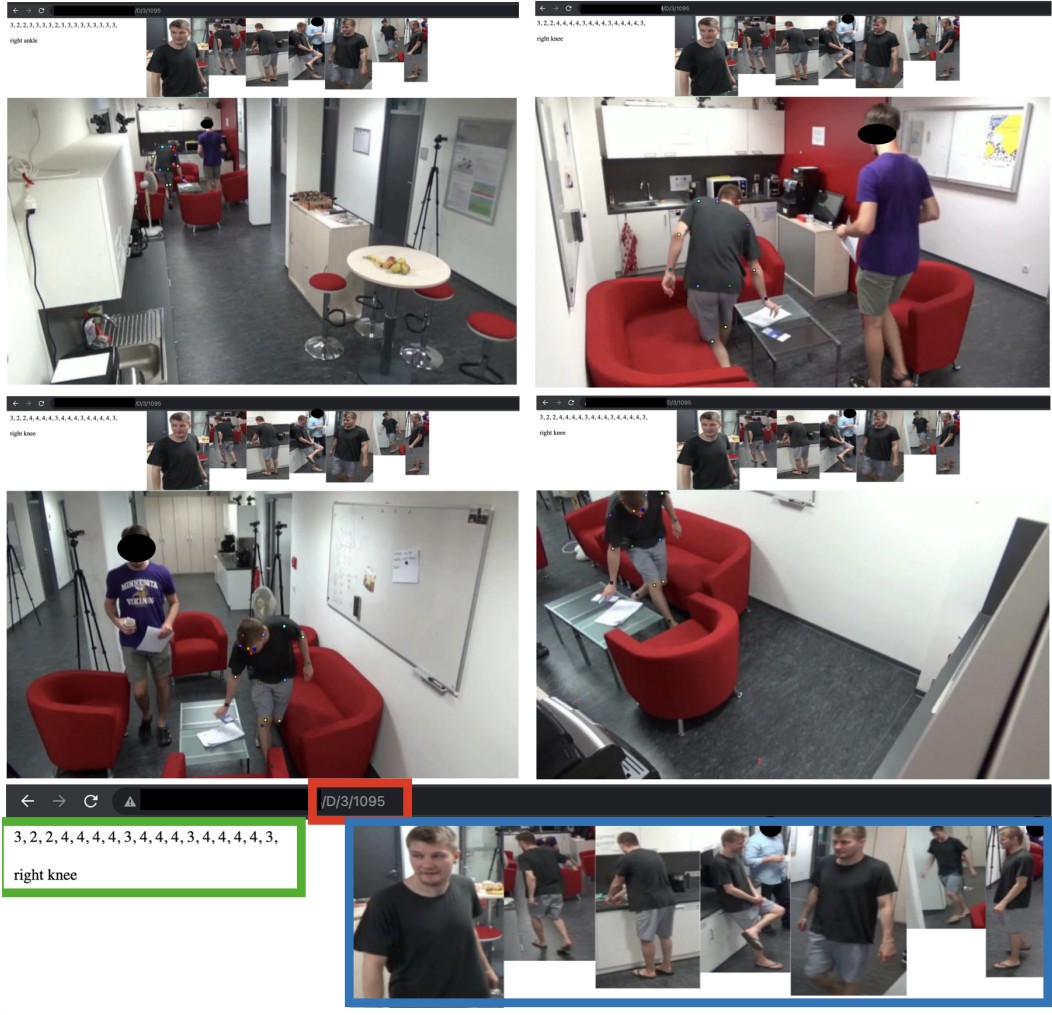

Figure 9: **3d pose annotation tool**: To correct the estimated 3d human poses, annotators can annotate a 17 keypoints OpenPose skeleton in all views using a custom annotation tool. The tool is browser based and allows accessing kitchen, person and frame via url (red box). The annotator is indicated which person they are supposed to annotate via a person banner (blue box). An annotation statistic informs the annotator about the amount of cameras in which they have annotated each joint (green box). For triangulation, each body joint has to be annotated in at least two camera views.

## 6 Baselines

### 6.1 Single-person Human Motion Forecasting

We conducted an evaluation of four recent single-person human motion forecasting methods: siMLPe [7], CHICO [8], pgbig [9], and History-Repeats-Itself (HistRep) [10], using the open-source codes provided by the respective authors. We adapt each method to fit our 29 joint skeleton but leave the hyper-parameters as is. During inference, we forecast in an auto-regressive way to produce sequences of arbitrary length. We found that normalizing the sequences such that the last input pose is at the origin and rotated such that the hip is on the x-axis improves the forecasting performance. In Table 1, we present the MPJPE difference between training on sequences in global coordinates and training on sequences after normalization. A positive value indicates that the normalized version of the method performs better while a negative value indicates that the global representation performs better. The impact of normalization varies across methods. CHICO [8] is not able to produce any reasonable forecast without normalization. siMLPe [7] also performs significantly worse without

| ΔMPJPE | 5 | 15 | 25 | 5 | 15 | 25 | 5 | 15 | 25 |
|---|---|---|---|---|---|---|---|---|---|
| | | walking | | | sitting down | | | whiteboard | |
| siMLPe [7] | 0.017 | 0.085 | 0.261 | 0.023 | 0.082 | 0.106 | 0.025 | 0.071 | 0.114 |
| CHICO [8] | 0.603 | 0.786 | 0.974 | 0.599 | 0.518 | 0.457 | 0.838 | 0.780 | 0.717 |
| pgbig [9] | 0.033 | 0.076 | 0.095 | 0.037 | 0.121 | 0.168 | 0.019 | 0.025 | 0.017 |
| HistRep [10] | 0.030 | 0.058 | **-0.034** | 0.011 | 0.070 | 0.090 | 0.025 | 0.045 | 0.031 |
| | | sink | | | cupboard | | | coffee | |
| siMLPe [7] | 0.024 | 0.052 | 0.050 | 0.021 | 0.035 | 0.041 | 0.024 | 0.069 | 0.277 |
| CHICO [8] | 0.531 | 0.454 | 0.374 | 0.527 | 0.494 | 0.420 | 0.528 | 0.568 | 0.634 |
| pgbig [9] | 0.017 | 0.038 | 0.036 | 0.016 | 0.053 | 0.081 | 0.013 | **-0.012** | 0.100 |
| HistRep [10] | 0.016 | 0.008 | 0.010 | 0.020 | 0.024 | 0.019 | 0.022 | 0.007 | **-0.051** |

Table 1: MPJPE difference between methods trained on sequences with and without global transformation, ↓ in dm. A positive value indicates that the normalized sequences perform better while a negative value indicates that methods trained on sequences without normalization perform better.

normalization. The impact of the normalization is lower for pgbig [9] and HistRep [10], but they also perform worse on average when the sequences are not normalized.

## 6.2 MRT

We use a PyTorch implementation of Multi-Range Transformers [11] provided by the authors. MRT is designed for a frame rate of 15 fps and 15 body joints, while Humans in Kitchens employs a frame rate of 25 fps and 21 body joints. We adapt the size of linear layers before and after the Transformer model to work with the different data dimensionality. We did not change the architecture or hyperparameters of the Transformer model itself.

We train MRT to predict three seconds of output given one second of input using 4818 overlapping multi-person samples from kitchens A, B, and C. Each sample consists of 1 to 16 people. We perform zero padding on samples with less than 16 people to ensure an equal number of persons throughout all samples, as in MRT [11]. We train for 50 epochs using adversarial training with the motion discriminator. We use a reduced batch size of 6 due to the higher number of persons per scene. When forecasting between one and three seconds, we perform auto-regressive predictions using the input sequence and all previous predictions as input for the Transformer model, as described by [11]. For long-term forecasting over three seconds, we only provide the last three seconds of forecast as input. We normalize each multi-person sequence to be centered around the origin. We calculate the mean of the x- and y-coordinate over all persons, joints, and frames and subtract it from each pose. This normalization removes differences in the global pose distribution between the kitchens A, B, C, and D. We found that MRT without normalization produces output sequences which are consistently shifted in comparison to the ground truth sequences, due to a different pose distribution between training and test data. We verify this in our experiments. Figure 10 shows that MRT performs worse without normalization.

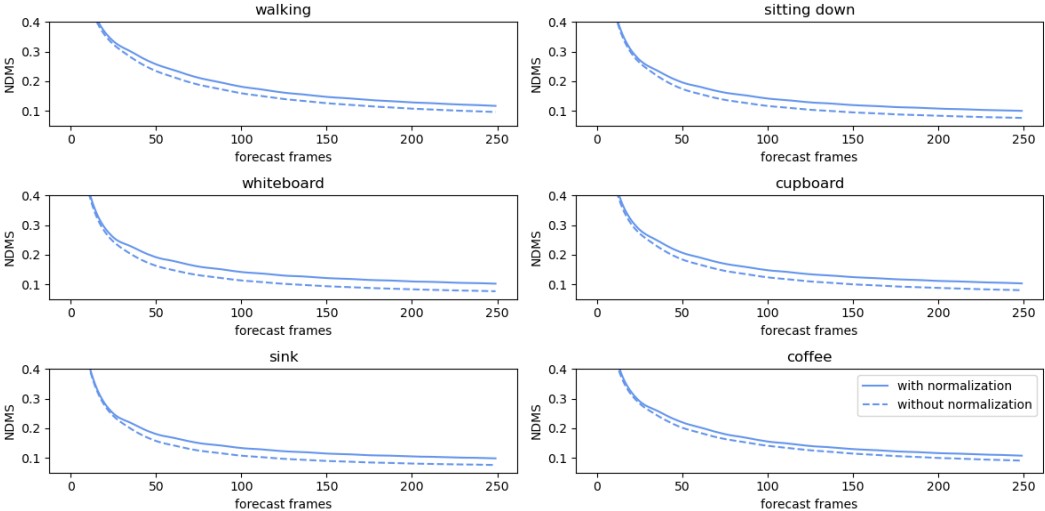

Figure 10: Without normalization, the ↑NDMS score is lower for longer time horizons.

| method | training data | 1s | 2s | 3s |
|---|---|---|---|---|
| [11] | [11] | 0.96 | 1.57 | 2.18 |
| [11] | [11] + Ours | **0.90** | **1.46** | **1.92** |

Table 2: MPJPE [dm] on [11] with and without extending the training data with our dataset.

| Protocol | hands | elbows | shoulders | feet | knees | hip |
|---|---|---|---|---|---|---|
| | | | 10 frames ($0.4s$) | | | |
| (a) | 0.024 (0.0087) | 0.018 (0.0077) | 0.010 (0.0038) | 0 | 0 | 0 |
| (b) | 0 | 0 | 0 | 0.016 (0.0040) | 0.014 (0.0071) | 0.008 (0.0053) |
| (c) | 0.024 (0.0088) | 0.014 (0.0074) | 0.010 (0.0036) | 0.015 (0.0041) | 0.013 (0.0065) | 0.008 (0.0051) |
| | | | 25 frames ($1s$) | | | |
| (a) | 0.041 (0.0188) | 0.026 (0.0112) | 0.014 (0.0057) | 0 | 0 | 0 |
| (b) | 0 | 0 | 0 | 0.039 (0.0158) | 0.033 (0.0196) | 0.010 (0.0072) |
| (c) | 0.047 (0.0206) | 0.027 (0.0114) | 0.014 (0.0057) | 0.036 (0.0142) | 0.032 (0.0196) | 0.010 (0.0064) |
| | | | 50 frames ($2s$) | | | |
| (a) | 0.069 (0.0323) | 0.041 (0.0178) | 0.015 (0.0070) | 0 0 | 0 | |
| (b) | 0 | 0 | 0 | 0.065 (0.0312) | 0.047 (0.0252) | 0.014 (0.0075) |
| (c) | 0.070 (0.0369) | 0.042 (0.0193) | 0.017 (0.0080) | 0.063 (0.0287) | 0.051 (0.0276) | 0.014 (0.0086) |
| | | | 100 frames ($4s$) | | | |
| (a) | 0.096 (0.0498) | 0.054 (0.0255) | 0.018 (0.0081) | 0 | 0 | 0 |
| (b) | 0 | 0 | 0 | 0.091 (0.0444) | 0.063 (0.0323) | 0.017 (0.0101) |
| (c) | 0.103 (0.0468) | 0.064 (0.0305) | 0.020 (0.0090) | 0.093 (0.0398) | 0.065 (0.0332) | 0.017 (0.0095) |
| | | | 1000 frames ($40s$) | | | |
| (a) | 0.190 (0.0947) | 0.087 (0.0349) | 0.023 (0.0097) | 0 | 0 | 0 |
| (b) | 0 | 0 0 | 0.087 (0.0379) | 0.060 (0.0270) | 0.019 (0.0090) | |
| (c) | 0.204 (0.1005) | 0.092 (0.0383) | 0.027 (0.0100) | 0.094 (0.0403) | 0.060 (0.0245) | 0.019 (0.0106) |

Table 3: Results for inpainting using MDM [12]. The three protocols denote (a) masking feet, knees and hip; (b) masking hands, elbows and shoulders; (c) masking upper and lower body, i.e., combining both (a) and (b). We report the average distance in [m] to the ground-truth joints for 16 samples, as well as the standard deviation in brackets.

### 6.2.1 Generalization to other Dataset

In Table 2 we replicate the experiment from [11] and extend their training data with our dataset. Using our dataset as additional training data reduces the error.

## 7 Diffusion Inpainting

We utilize the publicly available MDM [12] for inpainting in the joint space. We train the model on AMASS where we sub-sample the sequences from 60Hz to 25Hz to match our dataset framerate. We evaluate how well MDM performs for motion in-painting in Table 3. We have chosen 5 interpolation lengths, 10 frames (0.4s), 25 frames (1s), 50 frames (2s), 100 frames (4s) and as extreme case 1000 frames (40s). We select 5 starting frames for 3 individuals where the motion quality was verified by a human annotator. We start each interpolation at the same frame independent of the interpolation length. We follow three protocols: (a) masking feet, knees and hip; (b) masking hands, elbows and shoulders; (c) masking upper and lower body, i.e., combining both (a) and (b). We sample 16 samples and report the average distance in [m] to the ground-truth joints, as well as the standard deviation over all samples in brackets. As expected, the error is slightly larger when MDM has to inpaint both upper and lower body. However, the overwhelming number of inpainting tasks are similar to (a) as feet and knees are most likely to be occluded or truncated. The upper body is usually only for a short time occluded.

## 8 Comparison to GTA-IM

In this section, we discuss some of the limitations associated with synthetic datasets. In particular, we compare to GTA-IM [13] which is built on top of the GTA game engine. GTA is optimized for gaming and uses certain simplifications. The most relevant are:

- Motion and human poses are only weakly linked: The human poses are only *played back* for visual fidelity while the actual actor motion is represented as a simple 3d point. This sometimes causes a *walk-against-a-wall* effect when the person gets close to objects in the scene.

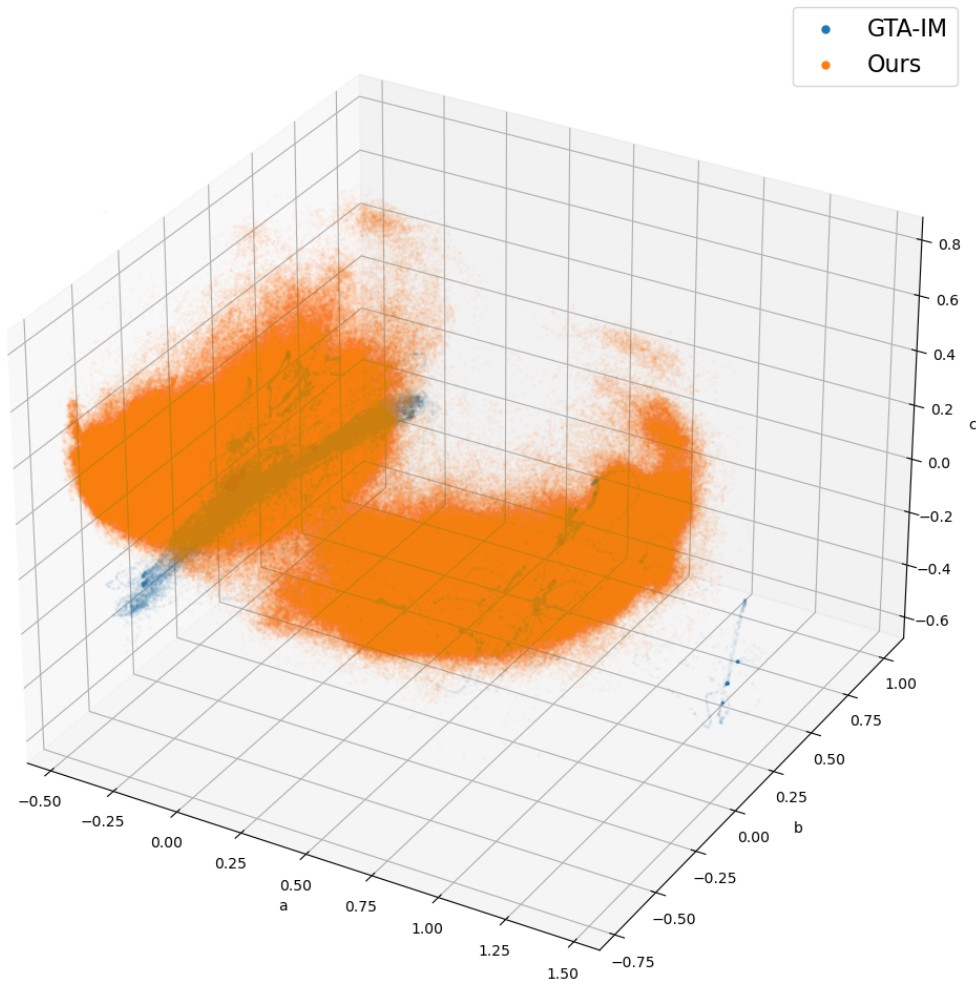

Figure 11: We project the 3d pose parameters of legs, normalized at the hip, and arms, normalized at the shoulder, of all poses in GTA-IM (blue) and in our dataset (orange) to a 3d space using PCA. We observe higher diversity in our dataset while GTA-IM forms an elongated cluster with a few outliers.

- Low motion diversity: game engines utilize a predetermined set of motion primitives while real humans express a wide range of motion variance. The motions of all persons are the same, while humans perform activities in different ways.

- Persons are not aware of the behavior of other actors. Collision are avoided using coarse bounding boxes.

In Figure 11, we compare the diversity of leg and arm poses between GTA-IM and our dataset. We observe higher diversity in our dataset while GTA-IM forms an elongated cluster with a few outliers. We further analyzed the GTA-IM dataset to quantify the differences between synthetic and real data for the same activity. As GTA-IM does not provide any action labels, we manually annotated "walking" motion for 92 minutes of the GTA-IM dataset, following the same annotation protocol as in our approach. We first evaluate how variable the walking speed is. In GTA-IM, the average walking speed is $1.4128 \frac{m}{s}$, which is the preferred walking speed[4], while the average walking speed is much lower ($0.6121 \frac{m}{s}$) in our indoor environment, as expected. More important, however, is the variation of speed, which is much lower for GTA-IM (std: $0.1642$) compared to our dataset (std: $0.2982$) captured in a real environment.

---

[4]https://en.wikipedia.org/wiki/Preferred_walking_speed

| dataset | Euclidean distance ↑ | variance ↑ |
|---|---|---|
| GTA-IM | 0.0420 | 0.0017 |
| Ours | **0.1563** | **0.0030** |

Table 4: Average distance of a pose of the walking sequence to their nearest neighbor.

Another common artifact of synthetic datasets is that similar pose patterns are repeated. We measure this by calculating the distance to the nearest neighbor of each pose in the walking sequence, utilizing the normalized poses. To compute the distance, we sample both datasets to 5Hz and then report the average distance of each pose in the walking sequences to their nearest neighbor in the entire walking sequence. The smaller this value is, the more similar are the motion styles. The results in Table 4 show that the similarity is much lower in our dataset.