# OpenReview forum: "Humans in Kitchens: A Dataset for Multi-Person Human Motion Forecasting with Scene Context"
_NeurIPS.cc/2023/Track/Datasets_and_Benchmarks — NeurIPS 2023 Datasets and Benchmarks Poster_

### Official Review · Reviewer_bd4f · 2023-07-18
**Review on Submission90**

**Rating:** 7
**Confidence:** 4
**Correctness:** The claims made are correct.

**Strengths:**

The provided data is very valuable, especially the per-frame activity label attached to 3D human instances and the dynamic scene geometry.
It would considerably facilitate the research on human-human/scene interactions.

With multiple off-the-shelf tools and considerable human labor, the annotation pipeline manages to provide fine-grained annotations.

Modelling scene geometry as boxes is a reasonable compromise.

**Additional Feedback:**

I appreciate the efforts in acquiring the dataset, also the potential of the provided dynamic scene geometry. I think the proposed database could help the community to better model complex human-scene interactions. Therefore, I'm giving a positive rating.

**Clarity:**

The paper is generally well-written, while some points are missed.
I'm curious on how the 3D object boxes are annotated. Are they acquired via custom tools?

**Documentation:**

The required information is provided.

**Ethics:**

There are no ethical concerns that warrant further discussion or review.

**Limitations:**

The authors discussed some of the limitations which I agree with.


**Opportunities For Improvement:**

The human pose annotation could be refined by utilizing more advanced techniques for key-point acquisition. Also, in the future I think it would be helpful to add hand and face poses to the current 3D pose data.

Current benchmark design could be improved, e.g., with tasks like Human-Object Interaction trajectory prediction to exploit the annotated dynamic scene geometry.

**Relation To Prior Work:**

There are some citations missed, like RICH[1], CHAIRS[2]. Discussion should be added.

[1] Huang C H P, Yi H, Höschle M, et al. Capturing and inferring dense full-body human-scene contact[C]//Proceedings of the IEEE/CVF Conference on Computer Vision and Pattern Recognition. 2022: 13274-13285.
[2] Jiang N, Liu T, Cao Z, et al. CHAIRS: Towards Full-Body Articulated Human-Object Interaction[J]. arXiv preprint arXiv:2212.10621, 2022.

**Summary And Contributions:**

The authors proposed Humans in Kitchens dataset to facilitate the research on complex relations between scenes and humans.
The dataset focuses on kitchen scenes, where natural daily human-scene interactions could happen.
For each scene, multiple humans are present, enabling multi-person interaction recording.
Furthermore, dynamic scene geometry with semantic labels is provided.
Finally, frame-wise human activity labels are also annotated.
Human motion forecasting is proposed as a first-step benchmark upon this dataset.

---

> ### Author Response · Authors · 2023-08-22
>
> **The human pose annotation could be refined by utilizing more advanced techniques for key-point acquisition**
>
> We use the pose estimation only as an initial estimate (Figure 3(c)). The 3D human poses are refined by human annotators (Figure 3(d)). A better pose estimation approach might have slightly reduced the annotation cost, but the used pose estimator scales well with the number of cameras and persons. The difficult cases are the frequent truncations and occlusions, where other methods struggle as well. Refining the manually corrected poses by another method would have introduced pose estimation errors.
>
> **It would be helpful to add hand and face poses to the current 3D pose data**
>
> This is an interesting future direction. While the camera resolution is 1024 x 900 pixels, the cameras capture large scenes. We are unsure how accurately hands and lip movements can be annotated.
>
> **Exploit the annotated dynamic scene geometry**
>
> We agree that this is an interesting direction. At the moment we do not have any results. Just concatenating the input with the distance to the moving objects did not improve the results.
>
> **3D object boxes annotation**
>
> The 3D object boxes were acquired with a custom tool, similar to Figure 9 in our supplementary material. We will provide more details of the tool in our revised supplementary material.
>
> **Related works RICH and CHAIRS**
>
> We will discuss RICH and CHAIRS in the revised version of the paper.
>
> Similar to PROX, RICH captures human-object interactions by defining pseudo-contact labels on the body mesh. Unlike PROX, RICH contains mostly outdoor scenes of around 60m^2 and provides more accurate SMPL-X estimates. Some of the scenes contain two humans who interact with each other from a distance (throwing a ball). Similar to other human-object datasets, the scene scans are static and do not change when people interact with it. In contrast, our dataset contains more people and the scene geometry changes.
>
> CHAIR is a large-scale human-chair interactions dataset that contains a large variation of chairs. Unlike our work, the dataset only contains chairs and is limited to a single person.

---

### Official Review · Reviewer_1cBr · 2023-07-20
**Review to Humans in Kitchens**

**Rating:** 6
**Confidence:** 4
**Clarity:** The overall paper is well written and…

**Strengths:**

1. The paper is well written, and dataset statistics and data collection details are clearly mentioned.
2. Multiple SOTA motion prediction methods for single and multiple individuals are evaluated. The authors report performance under two prediction settings, providing useful benchmarks for further work.
3. Addressing the existing gap in current motion datasets, the authors aim to capture more motions of interactive subjects and human-scene interactions. The incorporation of this large-scale dataset is envisioned to contribute to advancing research in this domain.

**Additional Feedback:**

Thanks for the reply.  The authors have addressed most of my concerns，and I would like to increase my rating.

**Correctness:**

I believe that the claims and dataset construction appear to be well-founded and reliable.

**Documentation:**

Information about the dataset, file formats, usage instructions, hosting plans, access, and licenses is detailed.

**Ethics:**

I do not see any obvious ethical concerns about this paper.

**Limitations:**

Apart from the aforementioned opportunities for improvement, another major limitation of this work is its insufficient generalization ability.
The authors themselves acknowledge the complexities of social and object-person interactions in various outdoor and indoor environments.
However, the dataset provided and the framework developed in this paper are limited to humans in kitchens, which severely restricts its applicability to other environmental scenarios. It is a significant limitation.

**Opportunities For Improvement:**

1. The available interaction types within the datasets are limited, primarily encompassing indoor activities such as walking, sitting, and engaging in object interactions such as catching or using items.
2. Qualitative results, particularly pose visualizations, are absent. I believe that intuitive visualizations play a crucial role in enhancing persuasiveness and understanding.
3. The geometry annotations are excessively coarse. While they do provide some spatial information, they lack the granularity required for human-object interactions.

**Relation To Prior Work:**

The related work section nicely discusses relevant datasets. The table comparing datasets seems comprehensive and helpful.

**Summary And Contributions:**

This paper introduces a new dataset (*Humans in Kitchens*) for multi-person motion prediction under scene context.  The dataset consists of 38 of the more than 4M unique poses of 90 individuals in total. The authors recorded people in four real kitchens for over 7.3 hours. Details on the acquisition and annotation of the dataset, dataset statistics, and evaluation of state-of-the-art methods for multi-person human motion forecasting are provided.

---

> ### Author Response · Authors · 2023-08-22
>
> **Interaction types within the datasets are limited**
>
> The goal of this dataset was not to record the most diverse motions, but to record humans interacting as naturally as possible. To facilitate this, we recorded in “natural” environments, i.e., in kitchens that are used by the participants on a daily basis, and provided minimal instructions to keep the behavior as unbiased as possible. The dataset is thus closer to an application than other datasets. For the evaluation, we sample the motion sequences to obtain a balanced evaluation for different activities. The evaluation is not dominated by walking or sitting. The diversity of the motion is actually even a little bit higher than for single person datasets. See general response (Diversity of motion) for a comparison to the single person dataset AMASS.
>
> **Qualitative results, particularly pose visualizations, are absent**
>
> We provided visualizations in the supplementary material and the provided dataset URL: https://drive.google.com/drive/folders/1cmR2M0lPhQsvGfT2aXg3WrY8Qvahzd9G
> In the folder “30s_clips”, we present various scenes from all 4 datasets for 30 seconds, where we rotate around the scene to get a better overview of the 3D geometry. In the folder “kitchen_dataset_as_video”, we show all 4 kitchen recordings uninterrupted from a fixed viewpoint.
>
> **Geometry annotations are excessively coarse**
>
> We have the scene geometry (see Figure https://drive.google.com/file/d/1AF9QhQLt5mNv3O2GyW8IwlUu4afj2WX0), but the focus is on learning based on the scene context where and how certain activities are performed. It includes sitting on a chair, drawing on a whiteboard, and using a coffee machine or microwave. We agree that the dataset cannot be used to model fine object interactions like rotating the knob of the microwave or grasping a knife, but it is a very useful dataset to model social interactions between persons and interactions with the scene. We will clarify this in the paper.
>
> **Limited to humans in kitchens**
>
> We agree that more scenes are desirable, but the four kitchens provide a rich environment with many activities. We furthermore train on 3 kitchens and evaluate on an unseen kitchen to measure the generalizability of the methods. We believe that the current dataset is unique as shown in Table 1 and a valuable source for multi-person human motion forecasting with scene context. The dataset can be complemented by other datasets recorded in different scenes in the future, as it has been the case for single person 3D human pose datasets.

---

### Official Review · Reviewer_Kghq · 2023-07-21
**Humans in Kitchens is valuably positioned to support forecasting the motion of multiple people in 3D.**

**Rating:** 7
**Confidence:** 3
**Clarity:** Yes, there are numerous typos but the…

**Strengths:**

Humans in Kitchens is a valuable new dataset for a number of reasons:

1. Existing datasets (outside of EgoBody) don't include multiple people and their 3D poses in larger scene coordinates.
2. Recent works like MDM show the promise of diffusion at learning generative models of human motion. However, those models generate motion in a vacuum, where the motion is only for a single person in an unknown 3D scene. Humans in Kitchens addresses an important *data gap* (multiple people in a scene) that existed for motion generation. Using Humans in Kitchens, one could now produce models that condition on multiple people when synthesizing a forecasted 3D motion.
3. At 7.33hr, this is quite a large dataset. I don't think I would consider it "large-scale" but the dense annotations definitely make it invaluable for the task of human motion forecasting you've targeted.
4. This dataset is relevant to the broader research community as methods in computer vision, reinforcement learning, and robotics can benefit from an understanding of multiple people moving in 3D. More specifically, understanding people as agents has been a compelling direction for these fields and although progress has been made in what can be seen, i.e., computer vision, progress is still lagging in terms of what embodied agents can accomplish and how they can move conditioned on their vision. Having this multi-person 3D keypoint dataset is valuable for furthering this effort, expanding opportunities for combining 3D keypoint motion synthesis with vision.
5. We had 3D keypoint datasets already in AMASS, and we have lots of 2D keypoint datasets with multiple people.. but having this Humans In Kitchens dataset fills in a missing blank while being real world enough to probably be useful.

**Additional Feedback:**

Typos/Suggestions:

24: onto the whiteboard -> on the whiteboard

38: four real kitchen -> four real kitchens

43: such was -> such as

48: we belief -> we believe

66: object -> objects

95: other hand is GDPR conform -> other hand conforms to GDPR

98: Persons were are allowed -> Persons were allowed

100:  Despite of the informed consent sheet (awkward)

133: activities where annotated -> activities were annotated

150: masked once -> masked ones

163: Is “MicrowaveKettle" two items i.e., a microwave and a kettle?

Fig 5: when persons perform -> they perform?

193: bird eye view -> bird's eye view

211: over very long time -> over a very long time

219: not for any other person -> not other people

219: This allows to report -> This allows us to report/This allows for reporting

220: still requires -> still requires us

228: approaches forecasts -> approaches forecast

238: As metric -> As a metric

268: We belief -> We believe

**Correctness:**

The dataset construction is sensible and encompassing. The forecasting benchmarks make sense as targeting a problem the community has not solved.

**Documentation:**

Yes, the supplemental and paper include URLs and the plan for maintaining and sharing the dataset.

**Ethics:**

No.

**Limitations:**

1. There are only a few hours of video in the dataset. These could have been recorded in a day and it is not really "large-scale". Compared to prior works it is large scale, and the quality and density of annotations is impressive (which is probably more important), but it's still only 7.33hr.
2. Pose estimation for many hours of people standing is not that interesting. Most existing 3D keypoint/motion datasets involve physical activities with lots of variation in pose. It's unclear that this dataset is sufficiently diverse to support modeling the motion of multiple people engaged in more interesting/varied activities.
3. As is, it feels like some additional analyses of the quality of their data are missing. Specifically, more analysis of how the new dataset differs from synthetic data and how existing methods fail on Humans in Kitchens would be valuable.

**Opportunities For Improvement:**

1. It would be nice if you had 10-20hr more of recorded RGB video data from your kitchens even if the data was unannotated (or weakly-annotated). Specifically many methods may try to learn from video and then evaluate against your carefully annotated poses. Although this is not specifically a direction you're targeting, it can be helpful to try to think of the bigger picture for how people might use data. Another nice thing about additional data from the same kitchens is that it might let methods further leverage the 3D boxes and cylinders you've annotated in some manner.
2. It would be worthwhile to do an assessment of the quality of the MDM fill-in for missing joints that you had masked. For example, if you randomly mask some joints you DO know about, how well does MDM fill-in those joints? This would be an opportunity for error bars and uncertainty measurements for masked-annotations-that-have-been-fixed-by-MDM.
3. It would be nice to have an analysis of where/when the existing forecasting methods fail. For example, a good justification of your dataset is how people's motion is influenced by the presence of others. So you could look at how existing motion forecasting baselines fail as a function of how many people are currently in the scene. These type of extra analyses really help motivate the value of the dataset.
4. It would be nice to analyze the limitations of synthetic data here. This was briefly discussed w.r.t the indoor GTA dataset, but given that the focus of the dataset is the 3D joints and not the RGB video, it would be worthwhile to quantify how the synthetic datasets (GTA included) fail at reasonably modeling 3D bodies. Specifically, lots of video game work on graphics/etc has gone into trying to model these things well, and it does seem like if it was modeled well it would obviously be a great source of 3D pose.

**Relation To Prior Work:**

Yes, the authors mention a number of existing related datasets and show how they are distinct from them.

**Summary And Contributions:**

Across 4 distinct kitchens, the authors use multiple cameras to film multiple people interacting, for between 1.5-2 hours each per kitchen. They then create a 3D human motion dataset from these captures. This human motion dataset consists of 3D OpenPose keypoints estimated per person, and person identity tracked manually. These 3D estimated keypoints are then corrected when/if the estimation failed, and masked if they are unsalvageable. Finally, after a 3D KP -> SMPL fitting, MDM conditional diffusion was used to infer the missing/masked keypoints. This dataset also includes 3D boxes and cylinders for objects that are in the kitchens (and including the walls). It does not appear that the authors release the RGB video used in the creation of this 3D dataset but it might have been released earlier and is not the main focus of their dataset.

The authors also provide two forecasting benchmarks. Forecasting as a task is appropriate for their multi-person dataset as a motivation for this work is understanding how people might move in the presence of other people.

---

> ### Author Response · Authors · 2023-08-22
>
> **It would be nice if you had 10-20hr more of recorded RGB video data from your kitchens even if the data was unannotated**
>
> We have one additional recording of Kitchen C with 12 cameras which is not annotated (12 x 2h). We can release the data as well. If there is a demand for additional data without annotations, we can record additional data in the four kitchens.
>
> **Quality of the MDM fill-in for missing joints**
>
> See general response.
>
> **Analysis of where/when the existing forecasting methods fail**
>
> See general response.
>
> **It would be nice to analyze the limitations of synthetic data**
>
> GTA is optimized for gaming and uses certain simplifications. The most relevant are:
> * Motion and human poses are only weakly linked: The human poses are only “played back” for visual fidelity while the actual actor motion is represented as a simple 3d point (“walk-against-a-wall” effect).
> * Low motion diversity: game engines utilize a predetermined set of motion primitives while real humans express a wide range of motion variance. The motions of all persons are the same, while humans perform activities in different ways.
> * Persons are not aware of the other actors behavior and avoid collision using coarse bounding boxes.
>
> In Figure https://drive.google.com/file/d/1dmRS5zFVSqjUt_MdA3VUdHnA36e0fcbs, we project the 3D pose parameters of legs (normalized at hip) and arms (normalized at shoulder) of all poses in GTA-IM (blue) and in our dataset (orange) to a 3D space using PCA. The plot shows a higher diversity in our dataset while GTA-IM forms an elongated cluster with a few outliers.
>
> **There are only a few hours of video in the dataset**
>
> The time-consuming part is not the recording, but the annotation effort. As shown in Table 1, even most single-person datasets are smaller. AMASS is larger, but it is a collection of existing datasets.
>
> **Diversity of motion**
>
> See general response.
>
> **Analysis of how the new dataset differs from synthetic data and how existing methods fail on Humans in Kitchens would be valuable**
>
> See response regarding synthetic data and general response regarding when existing forecasting methods fail.
>
> **Typos/Suggestions**
>
> Thank you for the detailed list. We will correct them.

---

> > ### Comment · Reviewer_Kghq · 2023-08-26
> >
> > Thank you for responding to some of the points I've raised. I think an analysis of how the proposed dataset includes a greater diversity of motion as compared to synthetic datasets like GTA would be quite interesting, ideally in line with the mentioned simplifications in your response. I think quantifying these differences is a valuable way to motivate the need for your proposed dataset.

---

> > > ### Author Response · Authors · 2023-08-28
> > >
> > > We further analyzed the GTA-IM dataset to quantify the differences between synthetic and real data for the same activity. As GTA-IM does not provide any action labels, we manually annotated “walking” motion for 92 minutes of the GTA-IM dataset, following the same annotation protocol as in our approach.
> > >
> > > We first evaluate how variable the walking speed is:
> > >
> > > | dataset | avg. walking speed | std. walking speed |
> > > | ----------|-----------------------------|--------------------------|
> > > |GTA-IM|  1.4128m/s                | 0.1642                   |
> > > |Ours     |  0.6121m/s                | 0.2982                   |
> > >
> > > GTA-IM follows the preferred walking speed (https://en.wikipedia.org/wiki/Preferred_walking_speed), while in our indoor environment the average walking speed is much lower as expected. More important, however, is the variation of speed, which is much lower for GTA-IM compared to our dataset captured in a real environment.
> > >
> > > Another common artifact of synthetic datasets is that similar pose patterns are repeated. We measure this by calculating the distance to the nearest neighbor of each pose in the walking sequence, utilizing the normalized poses described in our initial response. To compute the distance, we sample both datasets to 5Hz and then report the average distance of each pose in the walking sequences to their nearest neighbor in the entire walking sequence. The smaller this value is, the more similar are the motion styles.
> > >
> > > |dataset | mean eucl distance | variance |
> > > |-----------|--------------------------|--------------|
> > > |GTA-IM | 0.0420                    | 0.0017    |
> > > |Ours     | 0.1563                     | 0.0030   |
> > >
> > > The results show that our dataset has a much higher variability of poses.
> > >
> > > While we manually annotated the dataset, we observed several unrealistic artifacts in GTA-IM, which are typical for video games:
> > > * When an actor interacts with an object, especially a door or a chair, the actor is teleported to better match the pre-recorded object-person interaction, see example https://drive.google.com/file/d/1COI5WIiggKlckz3KBkX0-FZpoP3FfqgK.
> > > * Against-the-wall-walking: When an actor is blocked by an object, the actor still continues to perform walking motion while the position does not change, see example https://drive.google.com/file/d/1Xlipg9kLf8w3GoKsw-uP77MDIIO2swBV, https://drive.google.com/file/d/1eAXbO_nldhaYa5XJK_um5EP55ZZLjlyD

---

### Official Review · Reviewer_khgs · 2023-07-24
**Reviews for the Multi-Person in Kitchens Dataset**

**Rating:** 6
**Confidence:** 4
**Correctness:** Somewhat sound. Please see the above …
**Clarity:** Good.

**Strengths:**

1. This paper is easy to read and provides comprehensive quantitative and qualitative information about the dataset.
2. This dataset addresses an interesting and important direction for 3D multi-person scenes with interactions and occlusions. It provides a large-scale dataset with 3D human pose annotations and 3D boxes for objects. These useful data will benefit 3D human-object interaction and behavior understanding.
3. This work provides comprehensive visualization to understand and validate the data.

**Additional Feedback:**

1. The benchmark for short-term and long-term forecasting should be improved, including the selected models, proper metrics, and input-output length in order to highlight and validate the contribution and significance of the proposed dataset.
2. Table 4 and Table 5 should describe the numbers in the first row.
3. Based on the main actions of walking, motion forecasting can be one of the tasks used in this dataset but does not take advantage of the dataset. I suggest exploring more tasks to make the dataset useful and significant.
4. [Minor] The Line#148 uses SMPLify-X[25] to obtain SMPL. Does it use SMPLify instead of SMPLify-X?

**Documentation:**

Yes, this dataset URL is provided.

**Ethics:**

No.

**Limitations:**

This work discusses some limitations of the main paper. However, I think there are additional limitations that should be discussed, as shown in the above part (about the SMPLX annotations, multi-person motion forecasting models, pre-trained forecasting models to generalize well across datasets).

**Opportunities For Improvement:**

1. In the introduction, some descriptions may be overclaimed and out-of-scope in the proposed dataset. For example, the dataset is proposed for human motion forecasting as shown in the title. The introduction describes many "understanding and anticipating human motion within groups" and "capturing those complex social and object-person interactions" in the first two paragraphs. This part should shorten the scope and focus on the impact, previous models, and datasets on multi-person motion forecasting with occlusions and interactions. Moreover, if the data could help the understanding and capturing the object-person interactions, like writing on a whiteboard or making coffee, we could expect the annotations to contain hand meshes and object meshes with contacts instead of only coarse SMPL and 3D Box as the annotations.

2. The related work should discuss more "related" works and datasets. For instance, EgoBody emphasizes multi-person motion modeling and has interactions and occlusions, which should be discussed in the texts instead of only listed in Table 1. PROX should be compared with the proposed datasets. Some works should be considered in this part [1-6].

3. For the SMPL annotation, I have not found the descriptions for multi-view 3D pose estimation. How do we guarantee the annotation quality? Some works will project the 3D annotation on videos to see the overlap or they will validate the annotation algorithm on well-annotated datasets quantitatively. Why use MDM for inpainting? Do you try some naive interpolation methods, like linear interpolation or this method[7]? [7] compared with some generative models for frame inpainting, showing that even simple interpolation methods will obtain better performance.

4. As a motion forecasting dataset, my main concern is motion diversity and challenges. Compared with Human3.6M and AIST++[8], this dataset seems to have simple trajectories with diverse walking motions. What are the main challenges this dataset brings? Moreover, as a dataset with multi-person occlusions and interactions, the proposed benchmarks mainly use single-person motion forecasting models (e.g., HistRep), and in most cases, HistRep obtains the SOTA. These experimental settings and observations may harm the main contribution and significance of the dataset. Lastly, as a larger motion dataset, we expect it can train a basic forecasting model and generalize across different actions and datasets (e.g., on the mostly used Human3.6M or PROX).



Reference:

[1] Populating 3D Scenes by Learning Human-Scene Interaction, CVPR 2021

[2] PLACE: Proximity Learning of Articulation and Contact in 3D Environments, 3DV 2020

[3] Matterport3D: Learning from RGB-D data in indoor environments. 3DV 2017

[4] The Replica dataset: A digital replica of indoor spaces, arxiv 2019

[5] One-Stage 3D Whole-Body Mesh Recovery with Component Aware Transformer, CVPR 2023

[6] https://openxdlab.org.cn/details/RenBody

[7] DeciWatch: A Simple Baseline for 10× Efficient 2D and 3D Pose Estimation, ECCV 2022

[8] AI Choreographer: Music Conditioned 3D Dance Generation with AIST++, ICCV 2021

**Relation To Prior Work:**

Somewhat clear. More related work should be introduced and compared as suggested above.

**Summary And Contributions:**

This dataset proposes a large-scale multi-person human motion dataset with annotated 3D human poses, scene geometry, and activities per person and frame. Specifically, it covers over 7.3 hours of recorded data of up to 16 persons simultaneously in four kitchen scenes, with more than 4M annotated SMPL poses. The key components of this dataset contain the devices, behavior protocols, pose and activity annotation, manual nose annotations, automated human pose estimation, manual occlusion masking and human pose correction, SMPL fitting and inpainting, and scene annotation. The experiments were conducted on various state-of-the-art methods on the human motion short-term and long-term forecasting tasks.

---

> ### Author Response · Authors · 2023-08-25
>
> **Introduction / hand meshes & object meshes**: We will clarify in introduction that object-person interactions are coarse and not include fine-grained hand motion. We have scene geometry (https://drive.google.com/file/d/1AF9QhQLt5mNv3O2GyW8IwlUu4afj2WX0), but focus on learning based on scene context where and how activities are performed. It includes sitting on chair, drawing on whiteboard, and using coffee machine or microwave, but the dataset cannot be used to model fine object interactions like rotating the knob of the microwave or grasping a knife
>
> **Related works**:
> EgoBody & PROX are in Table 1. We discuss PROX briefly in related work. EgoBody is indeed not discussed in related work and we will include a discussion. We will discuss [1-2,6,8] in final version. We do not think that [3-5, 7] are related references
>
> **SMPL annotation**: As shown in Figure 3, we use multiple steps. We estimate multi-view 3D poses using [34] (l.135-139). The estimated 3D poses are manually corrected (l.140-147). The 3D poses are thus manually verified and corrected. We adjust the following off-the-shelf method for SMPL extraction https://github.com/Dou-Yiming/Pose_to_SMPL for our 3D skeletons where we keep the shape (betas) for each actor fixed. We determine shape for each actor beforehand as we are aware of their height, weight and general body circumferences. We use neutral body for every actor. We will clarify this in final version of the paper
>
> **Annotation quality**: Annotated 3D human poses are more accurate than the initially estimated 2D human poses, which contain many errors due to the large amount of people present in the scene. We therefore cannot use the projection as a quality measure. Our annotation process makes heavy use of “human in the loop” (Figure 3 b, d, e), which ensures high pose annotation quality when humans are visible from multiple cameras. If persons are occluded such that the annotators cannot annotate, we use linear interpolation or MDM. See general response for MDM
>
> **Why use MDM?**
> We use linear interpolation for occlusions for maximum 8 frames (1/3s). For longer occlusions, linear interpolation was not useful and we used MDM. [7] could have been used but we chose MDM since it is more recent, handles partial occlusions, and worked well on our data. Comparing different methods for motion inpainting is not the focus of this work. See general response regarding accuracy of MDM
>
> **H3.6M/AIST++**: our dataset contains natural motion in contrast to the pantomimed motion in H36M and dance motion in AIST++. We focus on interactions between persons and scene context. Neither Human3.6M nor AIST++ address interactions or context. See general response for diversity & challenges
>
> **single-person forecasting models**:
> The reason for lack of multi-person motion forecasting models is lack of large, realistic multi-person datasets. Our dataset will be essential to develop new multi-person motion forecasting approaches. MRT has been evaluated on very small or synthetic datasets and our results show that it does not outperform single person approaches on real data. The limitation of MRT is the normalization, which is not suitable for larger scenes, as they occur in our dataset. As discussed in Chapter 7.2 of the suppl. material, removing the normalization of MRT results in a drift. This issue can be addressed by future methods for multi-person motion forecasting
>
> **Generalize across different datasets**:
> Human3.6M and PROX are single-person datasets. We therefore demonstrate the generalization across datasets using the multi-person evaluation protocol that was used in [42] (MRT). We replicate the experiment from [42] and extend their training data with our dataset (MPJPE in [dm], lower is better):
>
> | method | training data          | 1s  | 2s   | 3s   |
> |---------|--------------------------|------|------|------|
> |MRT | [42] CMU               |0.96|1.57|2.18|
> |MRT | [42] CMU + ours    |0.90|1.46|1.92|
>
> This shows that our dataset generalizes very well across datasets
>
> **Additional limitations**: We will discuss additional limitation of dataset (SMPL annotations). See general response regarding imitations of models
>
> **The benchmark should be improved**: The selected models represent SOTA methods for single- and multi-person motion forecasting. The input and output lengths as well as the metrics are discussed in paper (L237-260)
>
> **Table 4 and 5**: The first row contains number of frames. We will clarify this.
>
> **More tasks**: As shown in https://drive.google.com/file/d/1ZqAVpUuRsKkP19NfGJxHGKVLy8jpvMSf, motion diversity is as high and for the head even higher compared to single-person dataset AMASS. The dataset can be used for multi-person motion forecasting, multi-person motion generation based on context and actions, multi-person pose estimation from a single view, and researchers will probably use it also for other tasks.The key aspect is **multi-person**
>
> **Line 148**: This is a mistake. We will correct it

---

### Official Review · Reviewer_zmcC · 2023-07-24
**Review for the well-annotated multi-human in kitchens motion dataset**

**Rating:** 6
**Confidence:** 4

**Strengths:**

1. This paper proposes a large-scale multi-person motion dataset containing more than 4M annotated human bodies and frame-wise activity for each person in 4 kitchen scenes. The recorded data involves up to 16 individuals at the same time within a single scene.

2. The authors provide a comprehensive description of the dataset's acquisition, annotation, and statistical analysis.

3. They thoroughly evaluate several state-of-the-art methods for the human motion forecasting task on the proposed dataset, comprising both short-term and long-term forecasting settings.

**Additional Feedback:**

1. Does your dataset provide scene geometry such as mesh or point cloud? Such a scene representation instead of 3D object box will be more useful for the task involving scene context setting.

2. What's the meaning of the descriptions in Line218 - Line219? Do you mean you only evluate the single person that doesn't perform the activity with other people.

Minor fix:

- $\mathbb{R}^{T-t \times n \times (29 \times 3)}$ -> $\mathbb{R}^{(T-t) \times n \times (29 \times 3)}$ in Eq. 1.

**Clarity:**

The paper is pretty well written and easy to read and understand.


**Correctness:**

The dataset construction process is systematic and the authors also provide a detailed procedure for data annotation. Additionally, the evaluation methods employed are straightforward and easy to comprehend.


**Documentation:**

The authors provide sufficient details.


**Limitations:**

The authors have discussed this paper's limitations and the societal impact.


**Opportunities For Improvement:**

1. The proposed dataset, with only four kitchen scenes lacking scene diversity, may constrain the model's ability to generalize to other scenes when forecasting human motion with scene context.

2. From my understanding, the dataset contains both single-person and multi-person activities, such as multi-person discussions. However, a statistical analysis of activities involving multiple persons is missing, e.g, the ratio of the multi-person activities in the dataset.

3. The authors emphasize that the proposed dataset will contribute to advancing the modeling of scene context for understanding social behavior and anticipation. Nonetheless, the experiments lack motion forecasting with scene context. A potential improvement could involve slightly modifying existing models by leveraging the 3D box information of objects as an additional input.

**Relation To Prior Work:**

The authors should consider discussing the following related works:

    - Zheng, Yang, et al. "Gimo: Gaze-informed human motion prediction in context." European Conference on Computer Vision. Cham: Springer Nature Switzerland, 2022.

    - Wang, Zan, et al. "Humanise: Language-conditioned human motion generation in 3d scenes." Advances in Neural Information Processing Systems 35 (2022): 14959-14971.

**Summary And Contributions:**

This paper introduces a new dataset called "Humans in Kitchens," a multi-person motion dataset encompassing over 7.3 hours of recorded data involving up to 16 individuals simultaneously in four kitchen scenes. The authors annotate more than 4 million human poses, per-frame dynamic scene geometry and object that people may interact with, and frame-wise activity for each person. Humans in Kitchens will contribute to advance multi-person human motion forecasting as well as modeling scene context for social behaviour understanding and anticipation.

---

> ### Author Response · Authors · 2023-08-22
>
> **Scene diversity**
>
> We agree that more scenes are desirable, but the four kitchens provide a rich environment with many activities. We furthermore train on 3 kitchens and evaluate on an unseen kitchen to measure the generalizability of the methods. We believe that the current dataset is unique as shown in Table 1 and a valuable source for multi-person human motion forecasting with scene context. The dataset can be complemented by other datasets recorded in different scenes in the future, as it has been the case for single person 3D human pose datasets.
>
> **Statistical analysis of activities involving multiple persons**
>
> In 27% of all frames we have activities that require at least two persons, like talking or writing on a whiteboard (subjects were asked to explain something to others). Furthermore, many activities are often induced by the actions of others, such as cutting cake or standing up from a chair. Figure 7 in our supplementary material shows an example where two persons are engaged in a conversation while also drinking coffee and eating fruits. Figure 8 of the supplementary material shows 3 actors being engaged by one person writing to the whiteboard. More examples are in our 30s clips of randomly sampled sequences from our dataset: https://drive.google.com/drive/folders/18HqC82gk0_iqwqlIZUtiFbVVwgY1VJ1c.
>
> **Leveraging the 3D box information of objects**
>
> We agree that this is an interesting direction. At the moment we do not have any results. Just concatenating the input with the distance to objects did not improve the results.
>
> **Related works GIMO and Humanise**
>
> We will discuss both works in our related works section and add them to Table 1.
>
> GIMO contains a single person interacting with a static, high-quality mesh, where data is provided in the form of an egocentric viewpoint. The dataset was explicitly designed for motion forecasting, where a person walks into a scene with the intent to interact with an object. Our dataset contains multiple persons, 5 times more frames, and moving objects.
>
> Humanise is a large-scale synthetic human-object interaction dataset that leverages existing datasets in human motion (AMASS) and 3D indoor scenes (ScanNet). Similar to GIMO, the scenes are static. In contrast to our work, Humanise is a single-person dataset. Furthermore, the person-object interactions are synthetic and do not include real person-object interactions.
>
> **Scene geometry such as mesh or point cloud**
>
> We have 3D scans for each of the 4 kitchens that have been captured prior to the recording session. See Figure https://drive.google.com/file/d/1AF9QhQLt5mNv3O2GyW8IwlUu4afj2WX0. It is important to note that this mesh is static and does not change over time, while the annotated objects can change their positions, e.g., chairs.
>
> **Line218 - Line219**
>
> In order to have a balanced evaluation across different activities, only the forecast motion of the actor performing the corresponding action is evaluated. The other persons present in the scene, however, impact the motion of the person and they are also used by the multi-person forecasting method.
>
> **Minor fix**
>
> Thank you. We will correct it.

---

### Author Response · Authors · 2023-08-22
**General Response**

**Analysis of when existing forecasting methods fail / Challenges** (khgs and Kghq)

We perform a qualitative analysis of MRT on samples with different numbers of persons per scene, to see how well a state-of-the-art multi-person model performs on our dataset with up to 16 persons interacting. In general, we find that MRT hardly generates sensible motion for more than 2-3 people. MRT generates good motions for dynamic activities, i.e., walking, sitting down or standing up. For motions which are more subtle, the model often freezes. This suggests that the joint generation of realistic motion of several people (>3) is challenging. Another major challenge is to handle varying types of motion like walking vs. writing on a whiteboard. We also find that MRT struggles to keep sensible distances between persons when a scene gets more crowded. For a high number of persons per scene (>8), we found multiple instances of a person walking into another person, see example videos (https://drive.google.com/file/d/1QOlhhxNhU2wqGRgXl5LW1sGIlrxvDP7b, https://drive.google.com/file/d/1P-R6zDZzFqh-NqqX6oyiv86RVrrzPw-_/view) This does not occur in scenes with fewer persons. This indicates that MRT is able to utilize information about the distances and relative locations between people, but its performance of doing so deteriorates with more people. We will add this discussion.

**Diversity of motion**  (khgs, Kghq, 1cBr)

We evaluated the pose variations and compared it to AMASS. Figure https://drive.google.com/file/d/1ZqAVpUuRsKkP19NfGJxHGKVLy8jpvMSf shows that for the leg and arm parameters of SMPL the variance is similar to AMASS and it is even higher for the head. The higher head movement is due to social interactions in our dataset where persons look at each other or at the objects they interact.

**Quantitative analysis of MDM** (khgs, Kghq)

We evaluate how well MDM performs for motion in-painting. We have chosen 5 interpolation lengths, 10 frames (0.4s), 25 frames (1s), 50 frames (2s), 100 frames (4s) and as extreme case 1000 frames (40s). We select 5 starting frames for 3 individuals where the motion quality was verified by a human annotator. We start each interpolation length at the same frame (for each of the 5 starting locations). We follow three protocols: (a) masking feet, knees and hip; (b) masking hands, elbows and shoulders; (c) masking upper and lower body, i.e. combining both (a) and (b). We sample 16 samples and report the average distance in [m] to the ground-truth joints, as well as the std. over all samples (in brackets).

10 frames (0.4s)
 |Protocol | hands | elbows | shoulders | feet | knees | hip |
 | --------|-------|--------|-----------|------|-------|------|
 | (a)     | 0.024 (0.0087) | 0.018 (0.0077) | 0.010 (0.0038) | 0 | 0 | 0 |
 | (b)     | 0 | 0 | 0 | 0.016 (0.0040) | 0.014 (0.0071) | 0.008 (0.0053) |
| (c)     | 0.024 (0.0088) | 0.014 (0.0074) | 0.010 (0.0036) | 0.015 (0.0041) | 0.013 (0.0065) | 0.008 (0.0051) |

25 frames (1s)
 |Protocol | hands | elbows | shoulders | feet | knees | hip |
 | --------|-------|--------|-----------|------|-------|------|
 | (a)     | 0.041 (0.0188) | 0.026 (0.0112) | 0.014 (0.0057) | 0 | 0 | 0 |
| (b)     | 0 | 0 | 0 | 0.039 (0.0158) | 0.033 (0.0196) | 0.010 (0.0072) |
 | (c)     | 0.047 (0.0206) | 0.027 (0.0114) | 0.014 (0.0057) | 0.036 (0.0142) | 0.032 (0.0196) | 0.010 (0.0064) |

 50 frames (2s)
 |Protocol | hands | elbows | shoulders | feet | knees | hip |
 | --------|-------|--------|-----------|------|-------|------|
| (a)     | 0.069 (0.0323) |  0.041 (0.0178) |  0.015 (0.0070) | 0 | 0 | 0 |
 | (b)     | 0 | 0 | 0 | 0.065 (0.0312) |   0.047 (0.0252) | 0.014 (0.0075) |
| (c)     | 0.070 (0.0369)  |0.042 (0.0193)  | 0.017 (0.0080)  |0.063 (0.0287) | 0.051 (0.0276)  | 0.014 (0.0086)  |

100 frames (4s)
 |Protocol | hands | elbows | shoulders | feet | knees | hip |
 | --------|-------|--------|-----------|------|-------|------|
 | (a)     | 0.096 (0.0498)  | 0.054 (0.0255)  |   0.018 (0.0081)  | 0 | 0 | 0 |
 | (b)     | 0 | 0 | 0 |  0.091 (0.0444) |0.063 (0.0323)| 0.017 (0.0101) |
 | (c)     |0.103 (0.0468) | 0.064 (0.0305) | 0.020 (0.0090)  |0.093 (0.0398)  |0.065 (0.0332)   |  0.017 (0.0095) |

 1000 frames (40s)
|Protocol | hands | elbows | shoulders | feet | knees | hip |
 | --------|-------|--------|-----------|------|-------|------|
| (a)    |  0.190 (0.0947) | 0.087 (0.0349)  | 0.023 (0.0097)  | 0 | 0 | 0 |
 | (b)     | 0 | 0 | 0 |  0.087 (0.0379) | 0.060 (0.0270)   | 0.019 (0.0090) |
 | (c)     | 0.204 (0.1005) | 0.092 (0.0383) | 0.027 (0.0100)  | 0.094 (0.0403)   |  0.060 (0.0245)  |  0.019 (0.0106)  |

As expected, the error is slightly larger when MDM has to in-paint both upper and lower body. However, the overwhelming number of inpainting tasks are similar to (a) as feets and knees are most likely to be occluded or truncated. The upper body is usually only for a short time occluded. We will include this analysis in the supplementary material.

---

### Decision · Program_Chairs · 2023-09-22

**Decision:**

Accept (Poster)

**Comment:**

This paper presents a dataset with multi-person human motions in scenes. The dataset is novel and interesting in its setting as it enables future research on human motion forecasting in scenes where multiple persons could be present. AC and all reviewers have reached a clear consensus regarding this point. The authors are encouraged to address the following weakness points in the final camera-ready version, however: 1) tone down some claims and rewrite the related paragraphs to make things clearer; 2) discuss a plan to add more videos in novel scenes or humans doing novel tasks; 3) present more thorough analysis and statistics for the dataset and the benchmarking experiments. AC and reviewers all agree on the acceptance recommendation of this paper.